

# Projected cryospheric and hydrological impacts of 21st century climate change in the Ötztal Alps (Austria) simulated using a physically based approach

Florian Hanzer[1], Kristian Förster[1], Johanna Nemec[2], and Ulrich Strasser[1]

[1]Institute of Geography, University of Innsbruck, Innsbruck, Austria
[2]ENVEO IT GmbH, Innsbruck, Austria

*Correspondence to:* Florian Hanzer (florian.hanzer@uibk.ac.at)

**Abstract.** A physically based hydroclimatological model (AMUNDSEN) is used to assess future climate change impacts on the cryosphere and hydrology of the Ötztal Alps (Austria) until 2100. The model is run in $100$ m spatial and $3$ h temporal resolution using in total 31 downscaled, bias-corrected, and temporally disaggregated EURO-CORDEX climate projections for the RCP2.6, RCP4.5, and RCP8.5 scenarios as forcing data. Changes in snow coverage, glacierization, and hydrological regimes are discussed both for a larger area encompassing the Ötztal Alps ($1850$ km$^2$, $862$–$3770$ m a.s.l.) as well as for seven catchments in the area with varying size ($11$–$165$ km$^2$) and glacierization ($24$–$77$ %).

Results show generally declining snow amounts with moderate decreases ($0$–$20$ % depending on the emission scenario) of mean annual snow water equivalent in high elevations ($> 2500$ m a.s.l.) until the end of the century, however decreases of $25$–$80$ % in elevations below $1500$ m a.s.l. Glaciers in the region will continue to retreat strongly, leaving only $4$–$20$ % of the initial (as of 2006) ice volume left by 2100. Total and summer (JJA) runoff will change little during the early 21st century (2011–2040) with simulated decreases (compared to 1997–2006) of up to $11$ % (total) and $13$ % (summer) depending on catchment and scenario, whereas runoff volumes decrease by up to $39$ % (total) and $47$ % (summer) towards the end of the century (2071–2100), accompanied by a shift in peak flows from July towards June.

## 1 Introduction

Current and future climate change is expected to significantly alter the mountain cryosphere of the European Alps. Rising temperatures are expected to result in more precipitation falling as rain rather than snow, a delayed onset of the snow-covered period, and an earlier onset of snowmelt, regardless of the considered emission scenarios (e. g., Frei et al., 2017; Gobiet et al., 2014; Marty et al., 2017). Possibly increasing winter precipitation amounts are expected to partly compensate for these temperature-induced effects of reduced average snowpack depths only in very high-elevated regions (e. g., Schmucki et al., 2015b).

The European mountain glaciers have already lost substantial parts of their volume and area during the past decades. Due to their delayed response to changed climatic conditions, they are still out of balance with the current climate and would continue to recede throughout the century even without any further climatic changes (e. g., Marzeion et al., 2014; Jouvet et al., 2011;





Zekollari et al., 2014). Further rising temperatures will only amplify this process, resulting in a total glacier volume reduction in the European Alps of 65–100 % by 2100 according to several global-scale studies (Bliss et al., 2014; Huss and Hock, 2015; Marzeion et al., 2012; Radić et al., 2013).

Consequently, the runoff regimes of snow- and ice melt-dominated Alpine catchments will also undergo significant changes. Meltwater contribution of the seasonal snow cover will be reduced, resulting in increased winter low flows, declining summer runoff, and shifts of the peak flows towards earlier periods of the year (Barnett et al., 2005; Horton et al., 2006; Stewart, 2009). These general projected patterns of change are of high confidence, as they are mainly triggered by increasing temperatures (shift from snowfall to rainfall, earlier onset of snowmelt), whereas in lower-elevated catchments changes in precipitation exhibit a larger impact on changes in runoff (Horton et al., 2006). Besides a projected future increase in winter precipitation on which most current climate projections agree on, future changes in precipitation over the Alpine region are however highly uncertain (Gobiet et al., 2014; Smiatek et al., 2016). Consequently, future trends in total annual streamflow volume for purely snowmelt-dominated catchments are also uncertain, as changes in precipitation might (over)compensate increased evaporation rates due to higher temperatures. While in glacierized catchments on the other hand runoff volumes will eventually decrease due to the retreating glaciers and subsequently reduced ice melt runoff volumes, prolonged periods of glacier mass loss can lead to increased glacier runoff volumes in the short- to midterm, depending on if increased melt rates are able to overcompensate the loss in glacier area (e. g., Jansson et al., 2003; Beniston, 2003; Collins, 2008; Bliss et al., 2014). Detecting the occurrence of this moment of peak water is of high interest, e. g., for hydropower production and planning (Schaefli, 2015).

While there are numerous studies on climate change impacts on the future of glaciers and hydrology for various parts of the European Alps, particularly Switzerland (e. g., Addor et al., 2014; Bosshard et al., 2013; Farinotti et al., 2011; Fatichi et al., 2015; Finger et al., 2012; Horton et al., 2006; Huss et al., 2008, 2014; Kobierska et al., 2013; Milano et al., 2015), few such studies exist for Austria. Kuhn and Batlogg (1998) used hypothetical temperature change scenarios to simulate future runoff for nine Austrian catchments with varying glacierization using a simple conceptual water balance model while assuming constant glacier areas. The same model was applied by Kuhn and Olefs (2007) for three catchments in the Ötztal Alps while accounting for changed glacier areas using an approach taking into account observed mean annual ice thickness changes with an additional constant surface lowering per degree of temperature increase. Tecklenburg et al. (2012) investigated climate change impacts on the Ötztaler Ache catchment using the conceptual semidistributed model HBV-D REG and one realization of the A1B climate scenario, however without accounting for glacier geometry changes but rather investigating only the two extremes of either constant glacier areas or entirely ice-free areas through the entire simulation period, respectively. Weber et al. (2010) used the fully distributed physically based model PROMET with the subgrid-scale extension SURGES for simulating glacier processes (Prasch et al., 2011) to calculate future hydrological changes in the Upper Danube basin using a single RCM realization based on the A1B scenario, and Wijngaard et al. (2016) applied the two conceptual hydrological models HBV and HQsim for the simulation of future hydrology in two catchments of the Ötztal Alps while updating glacier extents in 10-year intervals using precalculated ice thickness distributions.

Most of the cited studies rely on air temperature and precipitation as meteorological forcing data, applying simple temperature index methods for calculating snow and ice melt. However, the degree-day factors are calibrated for past conditions and





their transferability in space and time is uncertain. Several studies hence have pointed out that more physical methods should be favored over classical temperature index melt calculations in climate change impact studies (e. g., Farinotti et al., 2011; Huss et al., 2009; Radić et al., 2013; Viviroli et al., 2011). Some studies have for example applied enhanced temperature index methods that also take solar radiation into account for melt calculation (e. g., Addor et al., 2014; Bosshard et al., 2013; Fatichi

et al., 2015; Finger et al., 2012), addressing the fact that glacier melt rates are especially sensitive to variations in solar radiation (e. g., Huss et al., 2009; Ohmura et al., 2007). Only very few studies (e. g., Kobierska et al., 2013; Weber et al., 2010) however have applied full energy balance melt models for climate change impact assessment. While their superiority to more empirical methods is undisputed under the premise of in situ recordings of the required meteorological variables at the point scale, it remains challenging to provide adequate meteorological forcing data for their application in distributed mode. Nevertheless,

due to their physical basis energy balance models are in principle better suited to account for changed climatic conditions than conceptual models (e. g., Klemeš, 1990; Walter et al., 2005; Pomeroy et al., 2007).

In this study we apply the fully distributed energy and mass balance model AMUNDSEN (Strasser, 2008) to assess future climate change impacts on the cryosphere and hydrology of the Ötztal Alps (Austria). Downscaled, bias-corrected and temporally disaggregated EURO-CORDEX scenario simulations for the RCP2.6, RCP4.5 and RCP8.5 scenarios are used as climatic

forcing data. We analyze the simulation results with regard to changes in snow cover, glacier extent and volume, and hydrology, and discuss the associated uncertainties.

## 2 Study site and data

The study site (Fig. 1) is situated in the Ötztal Alps (Austria), a heavily glacierized Central-Alpine mountain range stretching east-west at the main ridge, with Austria in the north and Italy in the south. The Ötztal Alps are characterized by a warm and dry

climate, with average annual precipitation sums being as low as 660 mm at station Vent (1890 m a.s.l.). In the study, we focus on the headwaters of the three north-south trending valleys Kaunertal, Pitztal, and Ötztal (located from west to east in Fig. 1), which are tributaries to the Inn river in the north. Elevations in the study site range from 862 to 3770 m a.s.l., with a mean elevation of approx. 2400 m a.s.l. For the analysis of the cryospheric impacts in terms of changes in snow and glacierization, we take the entire area shown in Fig. 1 into account (1850 km$^2$), while the hydrological investigations are carried out for the

seven gauged catchments highlighted in the figure and listed in Table 1, with a total area of 379 km$^2$. The runoff regimes are characterized by a strong seasonality due to being fed mostly by snow and ice melt (glacial to nival regime types). Several of the catchments contribute to the Gepatsch hydropower reservoir situated in the south of the Kaunertal valley, whose natural catchment area of 106 km$^2$ is extended by a further 171 km$^2$ by diversion from the adjacent Pitztal and Radurschltal valleys. In the year 1997, 206 glaciers with a combined area of 150.7 km$^2$ were located in the area as determined from the second

Austrian glacier inventory (Kuhn et al., 2015).

For the model simulations, a digital elevation model (DEM) resampled to 100 m resolution was used. Initial ice thickness distributions for all glaciers in the region were calculated based on the glacier outlines and surface elevations of the year 1997 using the methodology by Huss and Farinotti (2012). The resulting ice thicknesses as shown in Fig. 1 were validated against



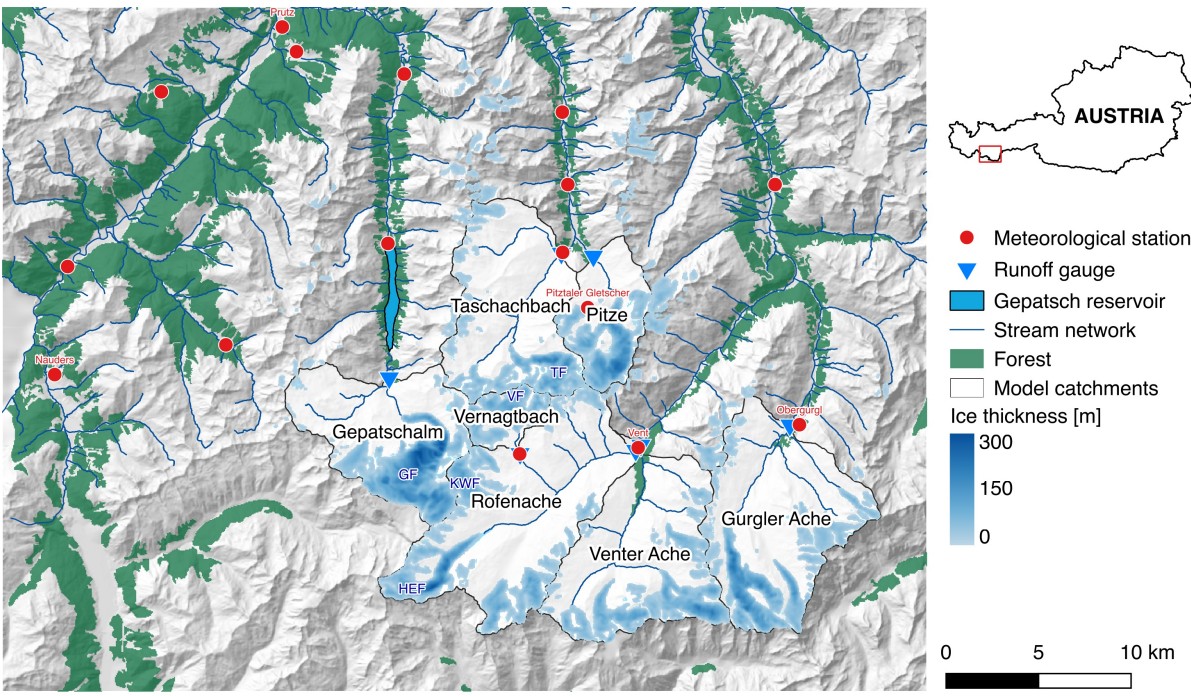

**Figure 1.** Location and topography of the study site including forested areas, catchment boundaries, the locations of the meteorological stations and runoff gauges, and the calculated initial ice thickness distribution for the year 1997. Selected meteorological stations and glaciers explicitly mentioned in the article are labeled in red and blue, respectively. Glacier abbreviations refer to Hintereisferner (HEF), Kesselwandferner (KWF), Vernagtferner (VF), Taschachferner (TF), and Gepatschferner (GF).

ground-penetrating radar measurements for eleven glaciers in the region, with deviations ranging between $-12$ and $26\,\%$ and an average deviation of $-3.2\,\%$ (Seiser et al., 2012).

Meteorological stations with long-term data records availability in the region were used to drive the model in 3-hourly temporal resolution for the historical simulations. For the scenario simulations until 2100, we used the EURO-CORDEX climate change projections (Jacob et al., 2013) as climatic forcing. The entire range of (as of July 2016) publicly available EURO-CORDEX simulations for the the high-resolution ($0.11° \approx 12.5\,\mathrm{km}$) EUR-11 domain that contained all required meteorological variables (minimum, maximum and mean $2\,\mathrm{m}$ air temperature, precipitation, relative or specific humidity, global radiation, and wind speed) in daily temporal resolution was selected, amounting to a total of 14 GCM-RCM combinations (Table 2). All of these 14 model combinations include simulations for both the RCP4.5 and RCP8.5 scenarios. In contrast to many other impact studies, we also included the intervention scenario RCP2.6, for which however only three realizations among the EURO-CORDEX pool were available. Hence, in total 31 different sets of climate projections were available for the glacio-hydrological simulations. The scenario period of the models is 2006–2100, with the exception of the three HadGEM-driven models which terminate in November 2099.





**Table 1.** Area, elevation, and glacierization (as of 1997) of the investigated catchments. The catchment IDs correspond to the labels in Fig. 1. Indentations indicate subcatchments.

| ID | Catchment | Area [km$^2$] | $z_{min}$ [m a.s.l.] | $z_{max}$ [m a.s.l.] | $z_{mean}$ [m a.s.l.] | Glacierization 1997 [%] |
|----|-----------|---------------|-----------|-----------|------------|-------------------------|
| 1 | Venter Ache | 165.3 | 1880 | 3762 | 2887 | 35.4 |
| 2 | — Rofenache | 98.6 | 1893 | 3762 | 2889 | 38.3 |
| 3 | —— Vernagtbach | 11.5 | 2638 | 3622 | 3120 | 77.0 |
| 4 | Gurgler Ache | 72.4 | 1885 | 3533 | 2785 | 31.8 |
| 5 | Gepatschalm | 53.9 | 1898 | 3524 | 2824 | 39.7 |
| 6 | Taschachbach | 60.5 | 1791 | 3761 | 2753 | 24.0 |
| 7 | Pitze | 27.0 | 1812 | 3548 | 2835 | 48.2 |
| | Total | 379.1 | 1791 | 3762 | 2833 | 34.4 |

**Table 2.** EURO-CORDEX scenario simulations used in this study.

| ID | RCM | GCM | RCPs |
|----|-----|-----|------|
| 1 | CCLM4-8-17 | CNRM-CM5 | 4.5, 8.5 |
| 2 | CCLM4-8-17 | EC-EARTH | 4.5, 8.5 |
| 3 | CCLM4-8-17 | HadGEM2-ES | 4.5, 8.5 |
| 4 | CCLM4-8-17 | MPI-ESM-LR | 4.5, 8.5 |
| 5 | HIRHAM5 | EC-EARTH | 2.6, 4.5, 8.5 |
| 6 | RACMO22E | EC-EARTH | 4.5, 8.5 |
| 7 | RACMO22E | HadGEM2-ES | 4.5, 8.5 |
| 8 | RCA4 | CNRM-CM5 | 4.5, 8.5 |
| 9 | RCA4 | EC-EARTH | 2.6, 4.5, 8.5 |
| 10 | RCA4 | CM5A-MR | 4.5, 8.5 |
| 11 | RCA4 | HadGEM2-ES | 4.5, 8.5 |
| 12 | RCA4 | MPI-ESM-LR | 4.5, 8.5 |
| 13 | REMO2009 | MPI-ESM-LR | 2.6, 4.5, 8.5 |
| 14 | WRF331F | CM5A-MR | 4.5, 8.5 |



## 3 Methods

### 3.1 Model

For the glaciohydrological simulations in this study, we used the fully distributed hydroclimatological model AMUNDSEN (Strasser, 2008). AMUNDSEN has specifically been designed as a scenario-capable model for the application in high-mountain

regions, and has been set up and extensively validated for historical conditions in the study site in a recent study (Hanzer et al., 2016). In the following, the most important model components are briefly discussed – for a more detailed model description, we refer to, e. g., Hanzer et al. (2014, 2016), Marke et al. (2015), Pellicciotti et al. (2005), Strasser (2004, 2008), and Strasser et al. (2008).

The model is capable of operating in temporal resolutions of 1–3 h, while the spatial resolution is generally arbitrary but

typically chosen in the order of 10–100 m for the application in high-mountain regions in order to accurately capture the complex topography and the underlying processes in these regions. As meteorological variables, AMUNDSEN requires point measurements or gridded inputs of air temperature, precipitation, relative humidity, global radiation, and wind speed. For the application using point data, the model includes a meteorological preprocessor for the spatial interpolation of scattered point measurements using a combination of lapse rates and inverse distance weighting. Lapse rates are either calculated from the

point measurements in each time step or are presupplied as average monthly values. A radiation model based on the method by Corripio (2002) is used to calculate potential clear-sky solar radiation while taking into account hill shading, transmission losses and gains due to scattering, absorption, and reflections, and uses measured point values of global radiation to derive cloud factor fields and subsequently actual global radiation fields. Similarly, incoming longwave radiation is calculated using parameterizations for radiation received from the clear sky, clouds, and surrounding terrain. The precipitation phase (rain

or snow) is determined as a function of the wet-bulb temperature. For solid precipitation, different correction methods are implemented in order to account for the undercatch of precipitation gauges when measuring snow accumulation. Hanzer et al. (2016) showed that a combination of a station-based snow correction factor (SCF) taking into account wind speed and air temperature with a subsequent constant post-interpolation SCF of 1.15 yielded plausible long-term precipitation amounts for the study area, while an additional redistribution of the interpolated snowfall fields using an approach based on topographic

openness (Yokoyama et al., 2002) distinctly improved the spatial snow accumulation patterns. Accumulated snow on the surface is subdivided into three layers called new snow, old snow, and firn. Transitions between new snow and old snow occur depending on snow density (which is calculated based on approaches by Anderson (1976) and Jordan (1991)), while remaining snow amounts at the end of the ablation season (September 30) are transferred to the firn layer. In glacierized catchments, an additional layer is used to track the evolution of ice amounts. A linear densification of the firn layer is employed, while firn is

converted to ice once reaching a threshold density of $900\,\mathrm{kg\,m^{-3}}$. Snow albedo is parameterized using an aging curve approach with exponential decay down to a specified minimum value, while firn and ice albedo is kept constant for this application (with values of 0.4 and 0.2, respectively). In forested areas the interpolated meteorological fields are modified in order to capture sub-canopy conditions, while additionally the effects of the forest snow processes of interception, sublimation, and melt unload are accounted for (Strasser et al., 2011). Snow and ice melt is calculated using an energy balance approach taking into account





short- and longwave radiation fluxes, latent and sensible heat fluxes, advective energy, as well as the ground heat flux. The delay of the onset of snowmelt in cold snowpacks is accounted for by a parameterization of the cold content of the snowpack, whereas melting snow may persist in the snowpack in the form of liquid water up to a certain amount (and possibly refreeze again) before actual outflow occurs (Hanzer et al., 2016). Evapotranspiration in snow-free areas is calculated following Allen

et al. (1998). Finally, runoff at predefined catchment outlets is calculated using a linear reservoir approach for snow on glaciers, firn, bare ice, snow outside of glaciers, and soil (Hanzer et al., 2016). Apart from the parameters of this linear reservoir model which have to be calibrated individually for each catchment, most parameters in the model have a physical meaning, and in general no site-specific calibration is performed.

The setup and extensive validation of the model for the study region under historical conditions has been described in Hanzer

et al. (2016). Essentially the same model setup was used for this study, aside from (i) the newly implemented glacier retreat parameterization as described in the following section, (ii) reducing the spatial and temporal model resolution from 50 m and 1 h to 100 m and 3 h due to performance reasons, and (iii) the fact that only a subset of the meteorological stations that were utilized in the simulations presented in the aforementioned study was available for this study due to the constraints of a sufficiently long measurement period required for the bias correction of the RCM data (see section 3.3).

## 3.2 Glacier geometry change

The model setup as described in Hanzer et al. (2016) already incorporated spatially distributed glacier thicknesses, however only accounted for the climatic forcing on the glaciers without any adjustment of glacier geometry. Due to the generally small vertical ice flow contribution for the majority of the glaciers in the study region (Helfricht et al., 2014) this is a reasonable approach for shorter simulation periods, however when performing simulations on multidecadal scales, the effect of ice flow

dynamics must be considered in glaciohydrological models. Otherwise, systematic errors in the simulated ice distributions (and subsequently in glacier runoff) are introduced as the glaciers thicken in their accumulation areas and retreat too quickly in the tongue parts.

For this study, we implemented the $\Delta h$ method (Huss et al., 2010) to periodically update the simulated glacier geometries. This approach is particularly suited for spatially distributed models operating on the regional scale, as it does not necessarily

require glacier-specific parameterizations but rather uses simple assumptions to translate the climatic forcing (i. e., the surface mass balance as computed by the mass balance model) into a geometric response in terms of the glacier thickness distribution.

Essentially, the $\Delta h$ parameterization scales the spatial distribution of the annual glacier surface mass balance such that the changes in glacier surface elevation match patterns observed in the past. This is accomplished by applying a prescribed function to each glacier in regular time intervals, which adjusts the simulated surface elevation change as a function of the normalized

glacier elevation (assumed to be a proxy for the central flowline). To derive the $\Delta h$ function, two glacier surface DEMs are required, preferably covering multidecadal periods to reduce possible errors due to DEM uncertainty.

In principle, a separate $\Delta h$ function could be derived for each glacier, however Huss et al. (2010) showed that glaciers with similar characteristics show similar geometric responses, and derived three $\Delta h$ functions for glaciers of different size classes: large valley glaciers ($A \geq 20\,\mathrm{km}^2$), medium valley glaciers ($5\,\mathrm{km}^2 \leq A < 20\,\mathrm{km}^2$), and small glaciers ($A < 5\,\mathrm{km}^2$):



$$\Delta h = \begin{cases} (h_r - 0.02)^6 + 0.12(h_r - 0.02) & A \geq 20 \text{ km}^2 \\ (h_r - 0.05)^4 + 0.19(h_r - 0.05) + 0.01 & 5 \text{ km}^2 \leq A < 20 \text{ km}^2 \\ (h_r - 0.30)^2 + 0.60(h_r - 0.30) + 0.09 & A < 5 \text{ km}^2, \end{cases} \tag{1}$$

where $h_r$ corresponds to the normalized surface elevation between 0 and 1.

While these parameterizations have been derived for the Swiss Alps, they are assumed to be applicable for all mountain glaciers (Huss et al., 2010), and have already been applied in other regions of the world (e. g., Huss and Hock, 2015; Ragettli et al., 2013). While for our study region DEMs and glacier outlines exist for the years 1997 and 2006, we chose to adopt these generalized parameterizations, as (i) they have been derived over much longer time periods, hence likely being more robust, and (ii) match the average observed $\Delta h$ patterns for our study area in the period 1997–2006 well (Fig. 2).

In AMUNDSEN, the glacier geometry update is performed at the end of each glaciological year (September 30): for each glacier, the total volume change in terms of the surface mass balance is used to scale the thickness change prescribed by the $\Delta h$ function (determined by Eq. 1) which is applied to each glacier pixel and the respective normalized elevations. Following Huss et al. (2010), the maximum surface lowering is limited to the surface mass balance lowering at the glacier terminus, and the glacier borders (pixels with ice thicknesses $< 10$ m) are not modified by the geometry update but rather change their thickness only due to the respective surface mass balance change.

Figure 3 demonstrates the effect of the glacier geometry update as implemented in AMUNDSEN exemplarily for the Hintereisferner and Taschachferner glaciers. While the average change in surface elevation between 1997 and 2006 is reproduced reasonably well for both glaciers in the run without updating the glacier geometries, the increase in surface elevation at the uppermost glacier parts is severely overestimated for both glaciers, whereas mass losses in the middle to lower parts of the glaciers are too high. Adding the glacier retreat module to the model distinctly improves the spatial distribution of mass gain or loss for both glaciers.

One limitation of the $\Delta h$ approach is that it allows only for geometric changes in terms of glacier retreat, essentially limiting its applicability for periods with largely negative mass balances. Huss and Hock (2015) present an extension of the method that enables to also account for advancing glaciers, however using a semidistributed model operating in elevation bands. A possible, comparatively simple extension of the method when applied in a fully distributed way could be to allow the glaciers to grow within their historical extents, similar to the approach applied by Stahl et al. (2008). In this study, however, we apply the original retreat-only version of the $\Delta h$ parameterization, which is seen a feasible approach given that future climate scenarios indicate climatic conditions causing intensive melt and, hence, glacier shrinkage.

### 3.3 Spatial downscaling of RCM data

While state-of-the-art RCM simulations already feature comparatively high spatial resolutions, usually the performance of the raw RCM data is still inadequate for directly using it in hydrological impact studies, both due to still-present mismatches in space as well as due to systematic model errors (biases) introduced by the RCMs. A common approach for impact studies is





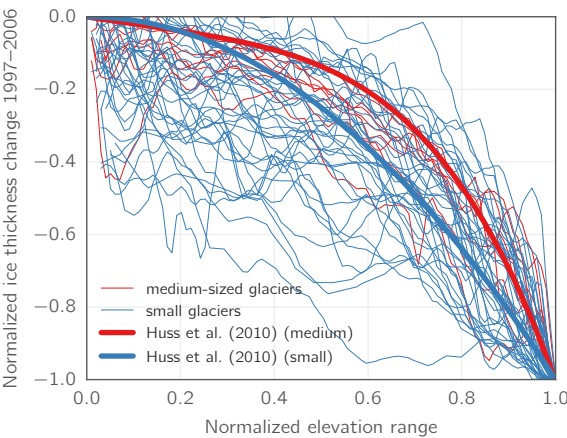

**Figure 2.** Normalized observed ice thickness change for 42 small ($A < 5$ km$^2$, thin blue lines) and 8 medium-sized ($5$ km$^2 \leq A < 20$ km$^2$, thin red lines) glaciers in the Ötztal Alps during the period 1997–2006, and the corresponding parameterizations from Huss et al. (2010) (thick lines).

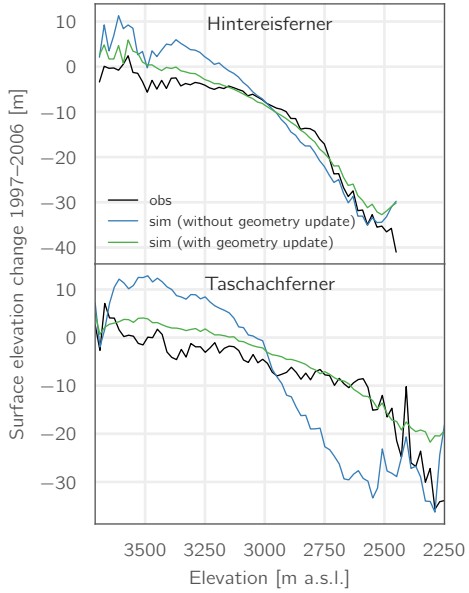

**Figure 3.** Observed and simulated glacier surface elevation change during the period 1997–2006 as a function of elevation for the glaciers Hintereisferner and Taschachferner (see Fig. 1 for their geographical setting). "Without geometry update" refers to the simulated surface mass balances without accounting for ice flow, while "with geometry update" refers to the simulation results obtained after implementing the $\Delta h$ parameterization.





hence to use ensembles of GCM-RCM chains in combination with statistial bias correction methods (Teutschbein and Seibert, 2012).

In this study, we use quantile mapping (QM) (e. g., Déqué, 2007; Themeßl et al., 2011a), an empirical bias correction method that corrects the distribution functions of the RCM variables to fit the distribution functions of the observations. QM has regularly shown to outperform other statistical bias correction methods (e. g., Teutschbein and Seibert, 2012, 2013), is applicable in mountain regions (e. g., Finger et al., 2012; Themeßl et al., 2011b, a), and allows for correcting climate variables other than temperature and precipitation (e. g., Finger et al., 2012; Wilcke et al., 2013). Like most statistical bias correction methods, QM does not explicitly account for spatial, temporal, or intervariable correlations, however it has shown to perform well under changed climatic conditions (Teutschbein and Seibert, 2013) and to retain intervariable relations (Wilcke et al., 2013).

In the study, we used the QM methodology by Gudmundsson et al. (2012) as implemented in the R package *qmap*. Due to strong season-dependent biases in the RCM data, we performed the bias correction separately for DJF, MAM, JJA, and SON. Only stations and variables with a minimum amount of 20 years of data within the period 1971–2005 were considered for the bias correction procedure, with the exception of global radiation where two stations with only approx. 10 years of data were also included due to an otherwise insufficient number of stations. Given these constraints, in total 16 precipitation time series were available, as well as 13 for air temperature, 6 for wind speed, 5 for relative humidity, and 3 for global radiation.

In order to avoid the possibility of unrealistic temperature values in terms of corrected $T_{min} > T_{max}$, following Thrasher et al. (2012) we did not correct $T_{min}$ and $T_{max}$ independently, but rather corrected $T_{max}$ and the diurnal temperature range (DTR) ($T_{max} - T_{min}$) with a subsequent calculation of corrected $T_{min}$.

For the EURO-CORDEX realizations for which only specific rather than relative humidity was available (IDs 1–4 and 13 in Table 2), mean daily specific humidity $q$ [kg kg$^{-1}$] was first converted to relative humidity RH [%] as

$$RH = 100 \frac{e}{e_s}, \tag{2}$$

with the vapor pressure $e$ [hPa] calculated as

$$e = \frac{qp}{0.622 + 0.378q}, \tag{3}$$

with $p$ being air pressure [hPa]. Saturated vapor pressure $e_s$ [hPa] was calculated following Sonntag (1990) as

$$e_s = \begin{cases} 6.112 \exp\left(\frac{17.62T}{243.12 + T}\right) & T \geq 0\,°C \\ 6.112 \exp\left(\frac{22.46T}{272.62 + T}\right) & T < 0\,°C \end{cases} \tag{4}$$

with $T = T_{mean}$ [°C].

With regard to the selection of RCM grid points for the downscaling to the point scale, we followed the approach by Hofer et al. (2017) (who however used linear regressions rather than QM) to find the *optimum scale* (OS) for each station and target variable: For each station and variable, spatial averages of the closest $1 \times 1, 2 \times 2, \ldots, 10 \times 10$ RCM grid points were calculated





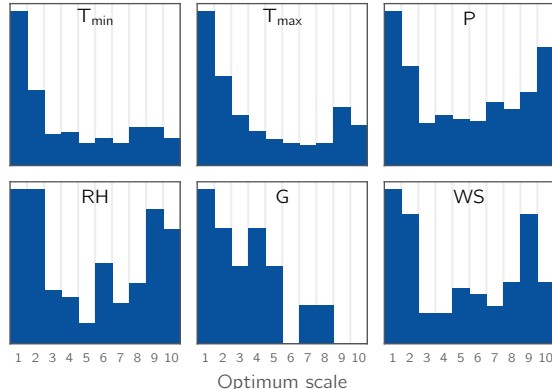

**Figure 4.** Histograms of the optimum scales for bias correction found using the approach by Hofer et al. (2017) for the variables minimum temperature ($T_{min}$), maximum temperature ($T_{max}$), precipitation (P), relative humidity (RH), global radiation (G), and wind speed (WS). Histograms are calculated using the derived optimum scales for all available stations for a given variable and all 14 GCM-RCM combinations (Table 2).

and subsequently used for bias correction. The OS for a given station and variable was then defined as the spatial window which minimizes the deviations between the cumulative distribution functions of the corrected and observed data in terms of the mean absolute error (MAE). Histograms of the resulting OS values for the different variables are shown in Fig. 4. While for all variables an OS of 1 (corresponding to using only the closest RCM grid point without spatial averaging) is the most common value, for all variables except global radiation all OS values from 1 to 10 are found – for precipitation, the maximum possible OS of 10 is even the second most common value.

RCM outputs from models using fixed 360- or 365-day calendars were linearly rescaled to Gregorian calendar dates (e. g., for a 360-day calendar each day from 1–360 was mapped to a respective day from 1–365 (or 366) in the Gregorian calendar). Resulting days with missing values were filled by duplicating the values from the preceding day.

Figure 5 shows the mean deviation (MD) and the standard deviation ratio (SDR) of the bias-corrected RCM data vs. the daily observations for the stations in the study area in the historical period 1971–2005. The model IDs (x-axis) correspond to the IDs in Table 2. While some models tend to perform slightly better than others on average, both measures are in general close to their optimum values (0 and 1, respectively) for all realizations, indicating that the corrected RCM outputs adequately represent the observed climate in terms of mean and variability. The higher deviations in the variability of relative humidity for some models are not a result of the bias correction per se, but rather of the required conversion of specific to relative humidity for these models.



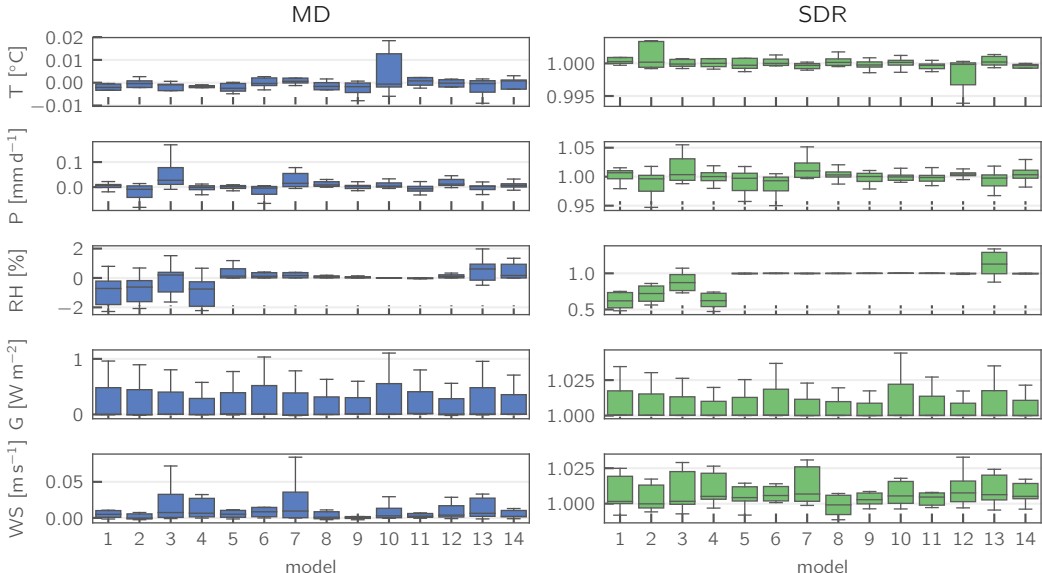

**Figure 5.** Mean deviation (MD) and standard deviation ratio (SDR) of simulated (i. e., downscaled and bias-corrected RCM data) vs. observed meteorological data for the historical period 1971–2005. The individual boxes represent the downscaled station meteorology for a given GCM/RCM combination (model IDs correspond to the ones in Table 2). Units given for the different variables apply only to MD (SDR is unitless).

## 3.4  Temporal downscaling of RCM data

As the EURO-CORDEX simulations were only available in daily temporal resolution, an additional processing step was necessary to derive the required sub-daily (1–3-hourly) data. This step was performed with the open-source temporal disaggregation tool MELODIST (Förster et al., 2016). MELODIST provides simple, empirical disaggregation routines for the sub-daily disaggregation of daily point-scale data for the variables air temperature, precipitation, relative humidity, solar radiation, and wind speed. For the application in this study several new disaggregation methods were added to MELODIST:

1. For air temperature, a disaggregation method conserving the daily mean temperature and the DTR ($T_{\max} - T_{\min}$) while applying a sinusoidal temperature course in the same way as the default method (which preserves $T_{\min}$ and $T_{\max}$ but not $T_{\mathrm{mean}}$) was implemented. However, most stations showed no clear sinusoidal temperature course, hence the application of this method still resulted in a (positive) temperature bias for most stations. Therefore, a new method based on mean values depending on the calendar month and hour of the day derived from hourly recordings was implemented. In a preprocessing step, for each month and station an average temperature course (normalized to a $[0, 1]$ range) is calculated. Then, daily values are disaggregated on the basis of these temperature courses again by either preserving $T_{\min}$ and $T_{\max}$, or $T_{\mathrm{mean}}$ and the DTR ($T_{\max} - T_{\min}$).



2. With regard to solar radiation, a similar method was implemented, which uses monthly varying average diurnal radiation courses derived from observations to scale the daily mean radiation. In addition, the methods for deriving daily radiation values from sunshine duration or the diurnal temperature range have been updated to allow monthly or seasonally varying conversion factors.

3. For relative humidity, an additional disaggregation method using [month, hour, dry/wet day] categorical mean values was implemented following Waichler and Wigmosta (2003). If daily humidity values are available, they can be used to scale the thereby derived values to preserve the daily mean.

Figure 6 shows the density functions for the observed hourly values (black lines) as well as for the various disaggregation methods exemplarily for station Obergurgl (1938 m a.s.l.) and the variables air temperature, relative humidity, global radiation,
and wind speed. Based on these comparisons and the results of the multilevel validation of the glaciohydrological simulation results when driven with the disaggregated values using multiple configurations, we decided for the following disaggregation methods: for all stations, air temperature was disaggregated using the "mean course" method while preserving daily minimum and maximum temperatures, daily precipitation amounts were distributed uniformly across the day, relative humidity was disaggregated using [month, hour, dry/wet day] categorical mean values while preserving daily mean radiation, solar radiation
was disaggregated using the "mean course" method while preserving daily mean radiation, and wind speed was distributed uniformly over the day.

Not all of these disaggregation methods performed best with regard to the reproduction of hourly values at the station scale, however this method combination yielded the best overall glaciohydrological model performance according to the multilevel validation procedure described in Hanzer et al. (2016). This is partly explainable from the fact that the disaggregation is
performed independently for each station and variable.

## 4  Results and discussion

In the following sections, first we discuss the future climatic changes in the study area as projected by the EURO-CORDEX scenarios. Then, we analyze the subsequent changes in snow, glaciers, and hydrology according to the AMUNDSEN simulations for the scenario period. These simulations were carried out by initializing the model in 1997 (using the glacier inventory
and ice thickness data) and running it until September 2006 using the observed meteorological data. Afterwards, we switched to the EURO-CORDEX scenario period and carried out the scenario simulations until 2100 (with the exception of the three HadGEM-driven models for which the simulations could only be carried out until the year 2098).

We did not perform the entire range of glaciohydrological simulations for a longer historical period due to restrictions of the current model setup: (i) the initial ice thickness distributions were calculated for the year 1997 and are hence not applicable
for earlier periods, and (ii) the generally negative mass balances of the glaciers in the Ötztal Alps during the last decades were interrupted by a short period of glacier advance between the mid-1970s and the early 1980s (Abermann et al., 2009), which cannot adequately be accounted for using the currently implemented $\Delta h$ parameterization for the glacier geometry update.





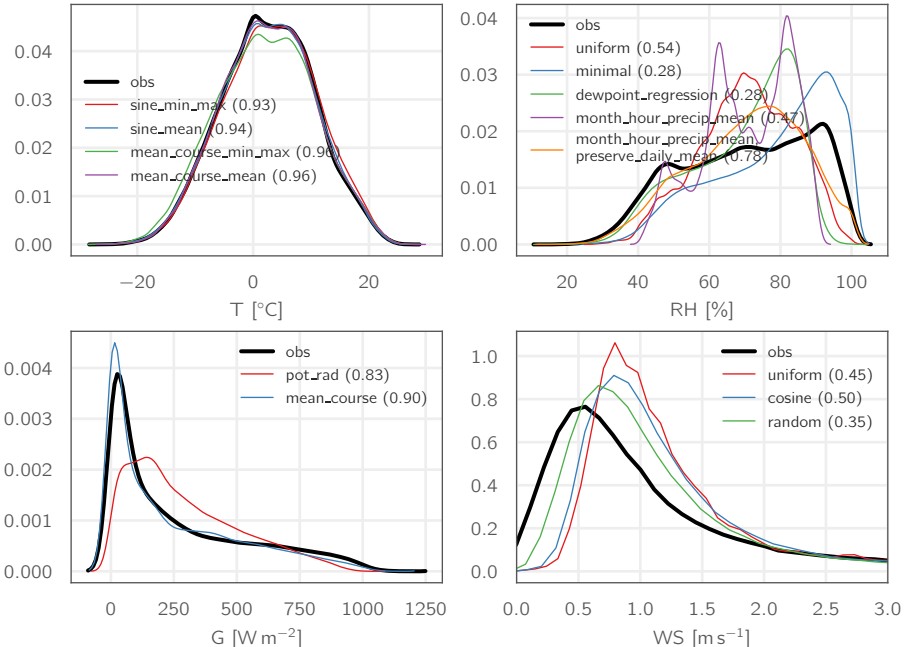

**Figure 6.** Density functions of the observed hourly values (black lines) for station Obergurgl (period 1999–2013) and the respective aggregated-and-disaggregated values using different disaggregation methods for the variables temperature, relative humidity, global radiation, and wind speed. Numbers in parentheses indicate the $R^2$ of the regression against the observations.

However, for discussing solely the changes in snow conditions, we also carried out model runs for the period 1971–2005 (hence obtaining transient runs from 1971 to 2100) using the historical RCM simulations as forcing, while neglecting the effect of glaciers in this case.

### 4.1 Future climate

5   Figure 7 shows the range of the projected seasonal and overall changes (spatial averages, relative to the baseline period 1971–2000) of the five meteorological variables for the study area as provided by the climate simulations. These changes are calculated based on areal means of the respective variables as calculated by the meteorological preprocessor in AMUNDSEN (i. e., after bias correction). While in the figure the changes are plotted for three time slices throughout the 21st century, for the following short discussion of the changes we focus on the period 2071–2100. If not otherwise stated, values refer to multi-model
10 means in the following.

    Temperature is projected to increase by all models and for all seasons, with average values of 1.1 (RCP2.6) to 3.8 °C (RCP8.5) in the annual mean, and a maximum spread of 2.2 °C between individual models for a given scenario. The highest warming is projected for the winter season, with up to 4.5 °C in the RCP8.5 scenario, while the smallest increases are projected for spring.





For precipitation, no clear general trend can be discerned with regard to annual sums. Both decreases (up to $-14$ %) and increases (up to 24 %) are projected by individual models, while the multi-model averages are close to zero for all three scenarios. However, with the exception of some outliers, there is a general consensus between the model towards a precipitation shift from summer towards winter. Projected multi-model average increases in winter precipitation are between 8 % (RCP2.6)

and 21 % (RCP8.5), with individual models projecting increases of up to 57 %.

Changes in relative humidity largely follow the trend of precipitation changes, with decreases in summer (up to $-1.5$ %) and smaller increases in winter (up to 0.7 %). At least for the RCP8.5 scenario slight decreases in the annual mean are projected.

For global radiation, the disagreement between the individual models is considerable. The multi-model means indicate a decrease in overall global radiation for all RCPs, with values between $-1.2$ W m$-2$ for RCP4.5 and $-2.4$ W m$-2$ for

RCP2.6, however with individual model results ranging between $-20$ W m$-2$ and 4 W m$-2$. The spread between the models is even larger when looking at seasonal changes. While all models agree on a decrease of global radiation in winter and (with one exception) spring, projections for summer are uncertain. Here, most models tend towards an increase in radiation for the RCP4.5 and RCP8.5 scenarios, however individual results range between $-29$ W m$-2$ and 26 W m$-2$.

Finally, wind speed projections also show quite a large spread between individual models. Multi-model means indicate a

very small overall decrease in wind speed, with seasonal increases of up to $0.12$ m s$^{-1}$ during winter and decreases of up to $-0.12$ m s$^{-1}$ during summer.

Comparing to other studies, the projected changes over the Ötztal Alps and their seasonal patterns are similar to the average changes projected for the entire Alpine region (Gobiet et al., 2014; Jacob et al., 2013; Smiatek et al., 2016). Notable differences are found for temperature, where the multi-model median change for 2071–2100 compared to 1971–2000 is $0.3\,°C$ lower than

in Jacob et al. (2013) for RCP4.5, and $1.0\,°C$ lower for RCP8.5. Seasonal temperature changes for the RCP4.5 scenario are between $0.1\,°C$ and $0.4\,°C$ lower than in Smiatek et al. (2016).

### 4.2 Changes in snow

Changes in the amount and duration of the simulated seasonal snow cover in the study area were analyzed both temporally and spatially. As a reference for computing the changes, we performed AMUNDSEN simulation runs using the historical periods

of the EURO-CORDEX simulations for the period 1970–2005. Through combination with the scenario runs we hence obtained transient simulations for the period 1970–2100.

Figure 8 shows the evolution of the simulated mean annual snow water equivalent (SWE) for three meteorological stations covering an approx. 2000 m elevation range: Prutz (870 m a.s.l.), Obergurgl (1938 m a.s.l.), and Pitztaler Gletscher (2864 m a.s.l.). At the highest-elevated station, Pitztaler Gletscher, the mean snow amounts remain relatively unchanged for all

three RCPs until the middle of the century, whereas only afterwards stronger decreases can be seen for the RCP4.5 and RCP8.5 scenarios. For RCP2.6, snow amounts are slowly increasing during the last third of the century, reaching an average level of 517 mm (multi-model mean), almost as high as the 530 mm in the historical period 1971–2000. However, also for the RCP4.5 and RCP8.5 scenarios, only comparatively small decreases in average SWE are simulated, amounting to 427 mm ($-18$ %) and 357 mm ($-31$ %), respectively. For station Obergurgl, similarly no strong changes in average snow amounts are simulated





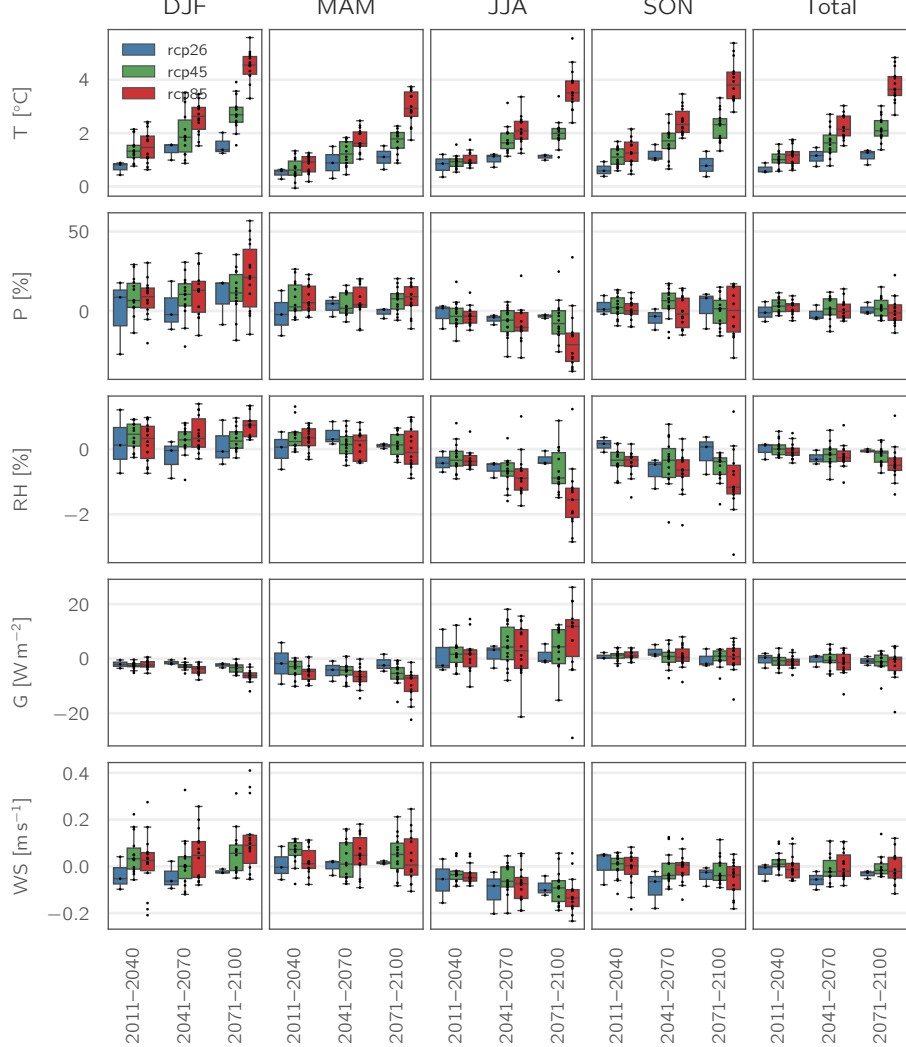

**Figure 7.** Projected seasonal changes for temperature (T), precipitation (P), relative humidity (RH), global radiation (G), and wind speed (WS) according to the selected EURO-CORDEX scenario simulations (spatial averages over the study area; changes calculated relative to 1971–2000). Small dots represent the individual GCM-RCM realizations.

within the first half of the century, whereas afterwards the curves of the three scenarios begin to diverge more strongly than for Pitztaler Gletscher. For RCP2.6 again a general trend of increasing SWE after 2050 is projected, while snow amounts in the RCP4.5 scenario decrease to 94 mm (−31 %) in the period 2071–2100, and 59 mm (−57 %) for the RCP8.5 scenario. The largest relative changes are found for the lowest-elevated station, Prutz, where the curves for the three scenarios begin to divert by around 2020. RCP2.6 snow amounts then stay relatively constant during the remainder of the century at 6 mm (−31 %),



while RCP4.5 and RCP8.5 snow amounts continue to decrease strongly, amounting to −63 % and −80 %, respectively, at the end of the century.

Similar results of stronger declining snow amounts for lower elevations can be seen in Fig. 9, where the multi-model mean change in SWE relative to the baseline period 1971–2000 is plotted against elevation. For RCP2.6, with the exception of the

very lowest-elevated parts of the study area, the strongest decreases in snow amounts generally are projected for the period 2041–2070, whereas the largest average snow amounts are found at the end of the century. Generally, however, the projected decreases are comparatively small, with maximum values of approx. 12 % for elevations above 2500 m a.s.l., up to 25 % for elevations between 1500 and 2500 m a.s.l., and slightly stronger below. For both RCP4.5 and RCP8.5, average snow amounts stay virtually constant (even slightly increasing for RCP4.5) during the period 2011–2040 in elevations above 2500 m a.s.l.,

whereas for the lower-elevated parts decreases during this period amount to up to 37 %. However, for both scenarios strong decreases are projected for the remainder of the century, with the largest changes in the lowest-elevated areas, amounting to up to −64 % and −83 % for RCP4.5 and RCP8.5, respectively.

The analysis of seasonal changes (Fig. 10) reveals that the strongest relative changes are projected for the summer months, while simulated changes for winter are comparatively small, amounting to maximum decreases of approx. 30 % for RCP8.5.

For all three scenarios, a gradual shift in the timing of peak SWE amounts from April towards March is simulated.

These projected changes in snow coverage generally show similar patterns to studies for other Alpine sites (e. g., Beniston et al., 2003; Marty et al., 2017; Schmucki et al., 2015a; Steger et al., 2012), however with a tendency to lower relative decreases in average snow amounts. This is likely due to the comparatively strong increases in winter precipitation especially for the warmest (RCP8.5) scenario (Fig. 7). A comprehensive study recently done for Switzerland (Marty et al., 2017) for

example projected more dramatic changes in snow amounts especially in high elevations (> 3000 m a.s.l.) until the end of the century for the A2 scenario (similar warming as in the RCP8.5 scenario), however projecting only very minor winter precipitation increases. Especially in high elevations however, increases in precipitation amounts are more likely to compensate for temperature increases, as temperatures in these elevations are usually still below melting conditions for most parts of the year.

### 4.3 Changes in glaciers

According to the simulation results, all glaciers in the study region will continue to retreat significantly throughout the 21st century. This is illustrated in Fig. 11, where the evolution of glacier volume and area (relative to the respective values of 2006) are plotted for all 31 model combinations as well as the respective multi-model means. In general, the evolution of glacier area and volume is very similar, with only a slightly stronger decline of glacier volume compared to the area. Looking at the multi-model mean values for the three emission scenarios, the largest changes in glacierization occur already within the first

half of the century, where – relatively independent of the emission scenario – the glaciers lose between approx. 60 % (RCP2.6) and 65 % (RCP8.5) of their volume from the beginning of the century. After 2050, the slope of the curves decreases and differences between the three scenarios become more prominent. In the RCP2.6 scenario, glacier volume decreases to approx. 25 % until 2080, whereas the glaciers stabilize afterwards and only slightly further retreat until 2100. In the RCP8.5 scenario, glaciers decrease more rapidly than in the RCP4.5 case between between 2050 and 2080, whereas afterwards the two curves





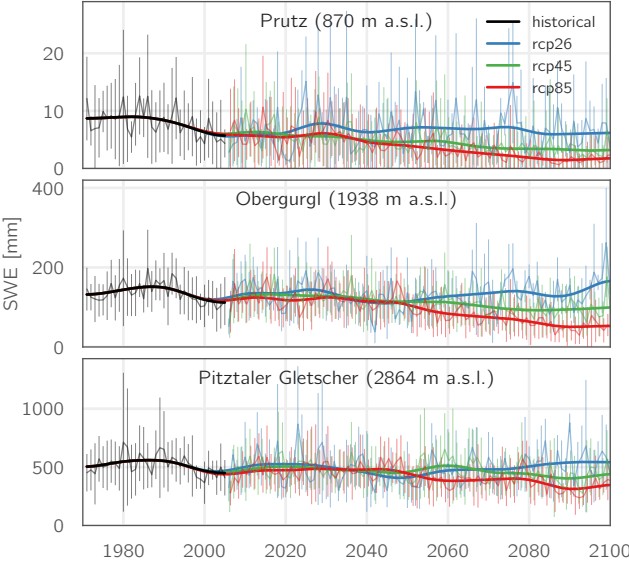

**Figure 8.** Evolution of mean annual SWE for the stations Prutz, Obergurgl, and Pitztaler Gletscher as simulated using the EURO-CORDEX scenarios for the RCPs 2.6, 4.5, and 8.5. Thin lines and error bars indicate the multi-model mean and standard deviation, respectively, while the thick lines show the 5-year Gaussian low-pass filtered multi-model mean.

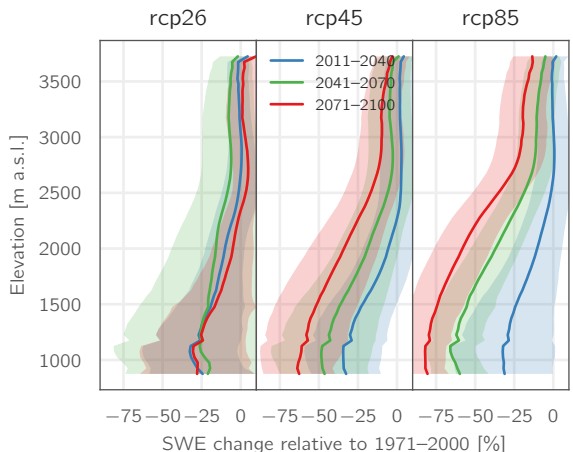

**Figure 9.** Elevation-dependent projected change in SWE (multi-model means, shadings indicate ± one standard deviation) relative to the period 1971–2000 for the RCPs 2.6, 4.5, and 8.5.

again follow a similar course. For both scenarios, the glaciers have mostly vanished by the end of the century, with only small



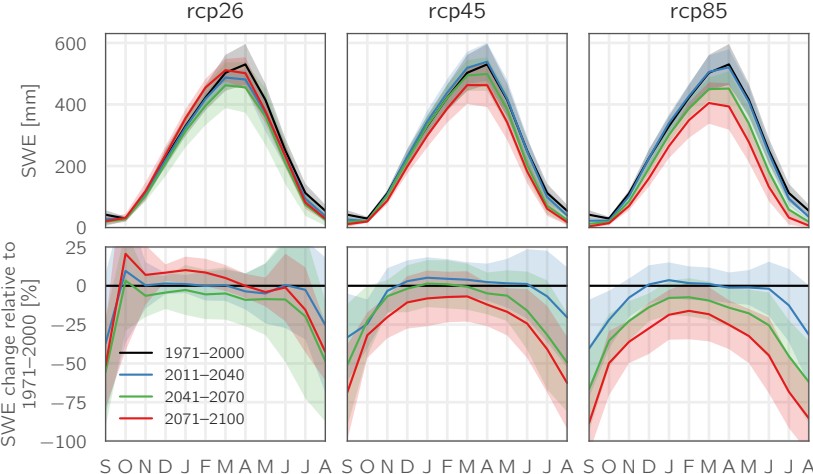

**Figure 10.** Mean monthly SWE (multi-model means, shadings indicate ± one standard deviation) over the entire study area (Fig. 1) for the three RCPs (top) and relative changes compared to 1971–2000 (bottom).

remains of approx. 10 % (RCP4.5) and 4 % (RCP8.5) still left in terms of their initial volume. Results for the evolution of glacier area are similar.

Looking beyond the multi-model mean, it becomes clear that there is a remarkable spread between the individual model runs, most strikingly for the RCP2.6 scenario, where the glacier volume decrease is between 58 % and 97 % until 2100 for the three model runs. Also for the RCP4.5 and RCP8.5 scenarios the range between the individual model realizations is considerable, with individual realizations resulting in practically ice-free conditions as early as 2070, while in other realizations up to 33 % of glacier volume are preserved until 2100.

The comparatively small influence of the different emission scenarios on the other side is visualized in Fig. 12, where the ice thickness evolution of a single glacier (Hintereisferner) along its centerline is plotted in 10-year intervals for a single GCM-RCM combination (REMO2009 driven by MPI-ESM-LR) and the RCPs 2.6, 4.5, and 8.5. Here again a strong loss of glacier volume can be seen in the first decades of the century. Regardless of the emission scenario, by 2040 the glacier retreats by approx. 1.4 km in length and loses approx. half of its volume (49–54 %). Only after 2060, the RCP8.5 scenario generates a markedly stronger loss in ice volume compared to the other two scenarios, resulting in an almost complete disappearance of the glacier by 2100. The results for RCP2.6 and RCP4.5 largely follow the same evolution until 2090. Only in the last decade of the century, the RCP4.5 scenario produces an accelerated retreat of the glacier, whereas the RCP2.6 glacier surface stays approximately constant. However, also for these two less extreme scenarios, only 25 % (RCP2.6) and 13 % (RCP4.5) of the original ice volume is left by the end of the century.

Figure 13 shows the evolution of the absolute and relative glacierization as well as the absolute glacier volume and mean ice thickness for each catchment in the RCP4.5 scenario, illustrating that different catchments partly show different responses




to the climatic forcing. For some of them (most notably Taschachbach) the average ice thickness stays approximately the same during the entire simulation period due to the similar rates of volume reduction and area reduction. For the Gepatschalm catchment, which includes the largest and thickest glacier in the study area (Gepatschferner), this is also the case for the first third of the century, whereas afterwards the rate of volume decline increases. In other catchments such as Vernagtbach and

Gurgler Ache the reverse effect is observed during the last part of the century: glacier area retreats more quickly than the volume decreases (visible by the increase in mean ice thickness in this period).

The spatial evolution of glacier coverage within the study area is shown in Fig. 14, where the simulated multi-model mean glacier coverage is displayed in 10-year intervals for the three emission scenarios. The rate of glacier retreat is mainly dependent on elevation and glacier thickness. In all three scenarios, most of today's small glaciers have disappeared by 2050 according

to the model. However, for the largest glaciers considerable differences resulting from the different emission scenarios appear. Gepatschferner, for example, loses comparatively moderate 54 % of its 1997 area (17.1 $km^2$) by 2100 in the RCP2.6 scenario, while in the RCP4.5 and RCP8.5 scenarios 71 % and 95 %, respectively, of the glacier area disappears.

## 4.4   Changes in hydrology

Figure 15 exemplarily shows the simulated seasonal runoff cycle for present-day conditions as well as the projected values

for the future periods for four of the study catchments. While the reaction of the catchments are slightly different depending on their characteristics, the general pattern of change is the same for all catchments and emission scenarios: summer runoff strongly decreases with simultaneously increasing spring runoff, indicating a shift from glacial/glacio-nival to nivo-glacial runoff regimes. While in the RCP2.6 scenario the month of peak runoff remains unchanged, in the RCP4.5 and RCP8.5 scenarios the peak gradually shifts from July towards June for all catchments, with the exception of Vernagtbach in the RCP4.5

scenario. For the Rofenache, Gepatschalm, and Pitze catchments, this shift already occurs in the period 2041–2070 in both scenarios. Only for Vernagtbach, the most glacierized catchment, in the RCP8.5 scenario the shift occurs only towards the end of the century. Monthly peak runoff slightly increases for the Pitze catchment in the period 2011–2040, however total annual runoff volumes stay approximately constant due to lower August runoff volumes. In the other three catchments, both monthly peak runoff and total annual runoff volumes do not exceed the levels simulated for the historical period in all scenarios. With

regard to ice runoff (dashed lines), Gepatschalm is the only catchment where ice runoff does not decrease monotonically over time, but rather increases in the period 2041–2070 compared to 2011–2040 before it strongly decreases towards the end of the century.

A more detailed analysis of the changes in monthly runoff is shown in Fig. 16, where the average monthly changes in runoff relative to the historical period 1998–2013 for the catchments in the study region throughout the 21st century are displayed.

With respect to the different emission scenarios, again the differences between them are very small within the first considered period (2011–2040). In this period, also no clear trend of changes in the seasonality of the runoff regimes can be discerned, apart for slight increases during the winter months and slight decreases during summer. In the following period, 2041–2071, a clearer pattern begins to emerge. While May to June runoff stays approximately unchanged, runoff strongly decreases during August and September in all three scenarios, amounting to up to −55 %. From October to November, a transition period





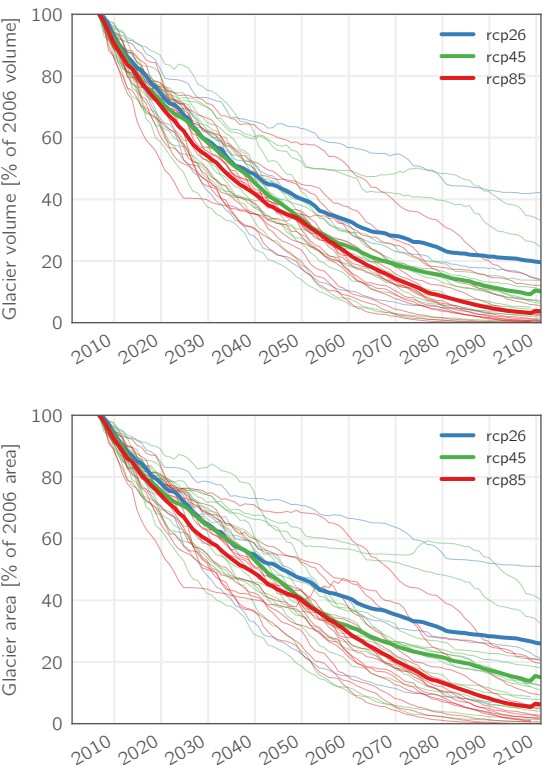

**Figure 11.** Simulated total glacier volume (top) and area (bottom, both shown as changes relative to the year 2006) for all glaciers in the study area and all 31 available climate scenarios. Thin lines indicate individual model results, thick lines show the multi-model mean. The small spikes in the multi-model means for RCP4.5 and RCP8.5 at the very end of the century are due to the fact that three models could only be run until 2098.

.

between decreasing and increasing runoff occurs, while during winter and spring increases of up to 50–125 % (depending on emission scenario) are projected (although still amounting to very small absolute values in comparison to summer as can be seen in Fig. 15). During the last part of the century (2071–2100), summer runoff continues to decrease, although not as strongly as between the first two periods. Winter runoff also increases, most notably for the RCP8.5 scenario, where November

5   to March runoff is at least approximately doubled for all catchments compared to the reference period. In this period, also different catchment characteristics can be more clearly distinguished. The largest changes are simulated for Vernagtbach, the smallest, highest-elevated, and (initially) most glacierized catchment. Here, increases between 43 % and 225 % for November to May runoff are simulated for the RCP8.5 scenario, while the corresponding runoff decreases by 70–75 % during August and September due to the by then almost completely melted glacier (Vernagtferner). However, while in the figure the relative

10   (positive) changes during winter appear similar or larger than the (negative) changes during summer for all catchments, it

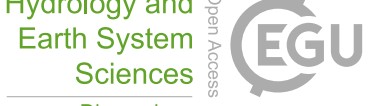



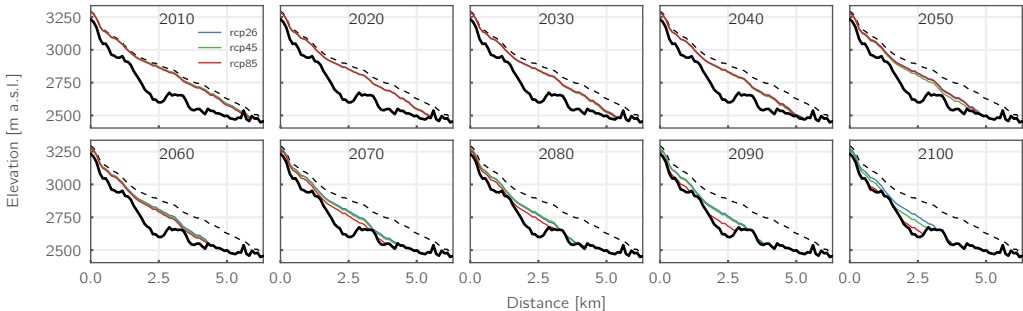

**Figure 12.** Bedrock topography (thick black lines) and simulated glacier thickness (thin colored lines) along the centerline of Hintereisferner as simulated for a single GCM-RCM combination (REMO2009 driven by MPI-ESM-LR) and the three RCPs, shown in 10-year intervals. Dashed black lines indicate the initial ice thickness as of the year 1997.

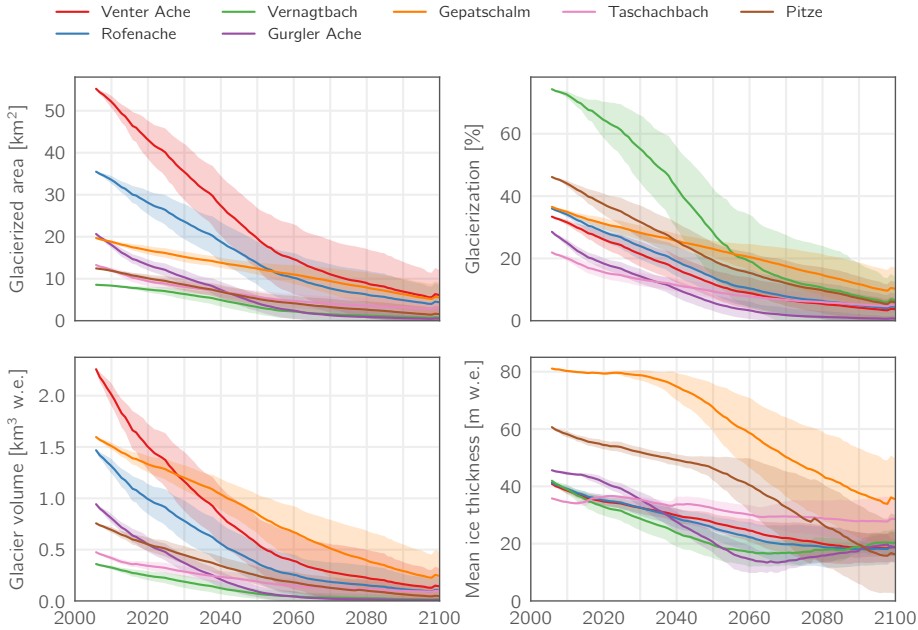

**Figure 13.** Evolution of the absolute (top left) and relative (top right) glacierized area for each catchment in the RCP4.5 scenario, as well as the absolute glacier volume (bottom left) and the mean ice thickness (bottom right). Note that the latter is calculated only over the glacierized fraction of the catchment and not the entire catchment area (i. e., it is the result of dividing the bottom left plot by the top left plot). Lines and shadings represent multi-model means ± one standard deviation.

has to be emphasized that this still corresponds to a strong relative decrease in total annual runoff considering the respective




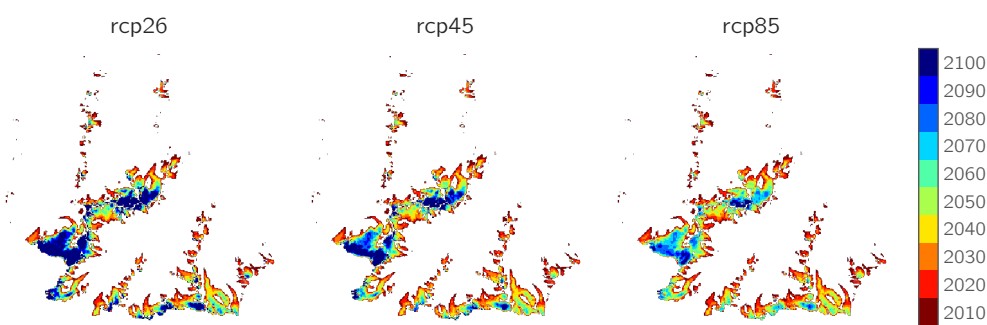

**Figure 14.** Spatially distributed simulated glacier coverage (multi-model mean) shown in 10-year intervals for the three RCPs.

absolute values (for example as seen in Fig. 15, the absolute JJA runoff volumes are approx. 2 orders of magnitude larger than the respective DJF values).

The simulated future streamflow composition of the components ice melt, snowmelt and rainfall at the end of the scenario period (2071–2100) is shown in Fig. 17. While the ice melt fractions at the end of the century are negligible for all but the most glacierized catchments, snowmelt is still the major contributor to total runoff for all seasons. Rainfall runoff contributions are comparatively low in general, which is however partly due to the model structure – as rain falling on snow contributes to the liquid water storage of the snowpack up to a certain amount, these rainfall amounts are part of the snowmelt contribution in the runoff concentration scheme. Only rain falling on bare ground, ice, firn or already saturated snow is part of the rainfall runoff contribution in this case.

## 4.5 Uncertainty

Huss et al. (2014) assessed the influence of various model assumptions in projections of glacier evolution and runoff in high-mountain catchments. Their results indicate that major uncertainty sources are especially (i) the winter snow accumulation in terms of both volume and spatial distribution, (ii) the approach to account for glacier geometry changes, (iii) the initial glacier ice volume distribution, and (iv) the individual climate projections.

While the model has shown to capture (i) the amount and distribution of winter snow accumulation well for the study area in a recent study (Hanzer et al., 2016), the model setup was slightly altered for the present study. The most substantial changes result from the requirement of a sufficiently long historical measurement period of the meteorological stations. As a consequence of this, only a subset of the stations utilized in the aforementioned study could be used, and from these stations only daily instead of hourly recordings were available, requiring an additional processing step in terms of temporal disaggregation. In addition, at several of these stations the daily and (sub-)hourly measurements are performed with different instruments, resulting in considerable differences in the recorded meteorological variables as illustrated in Fig. 18 for three stations. Especially the latter impacted model performance, resulting in a tendency to slightly overestimate winter snow accumulation and spring runoff volumes. Reducing the spatial and temporal model resolution from 50 m and 1 h to 100 m and 3 h on the other





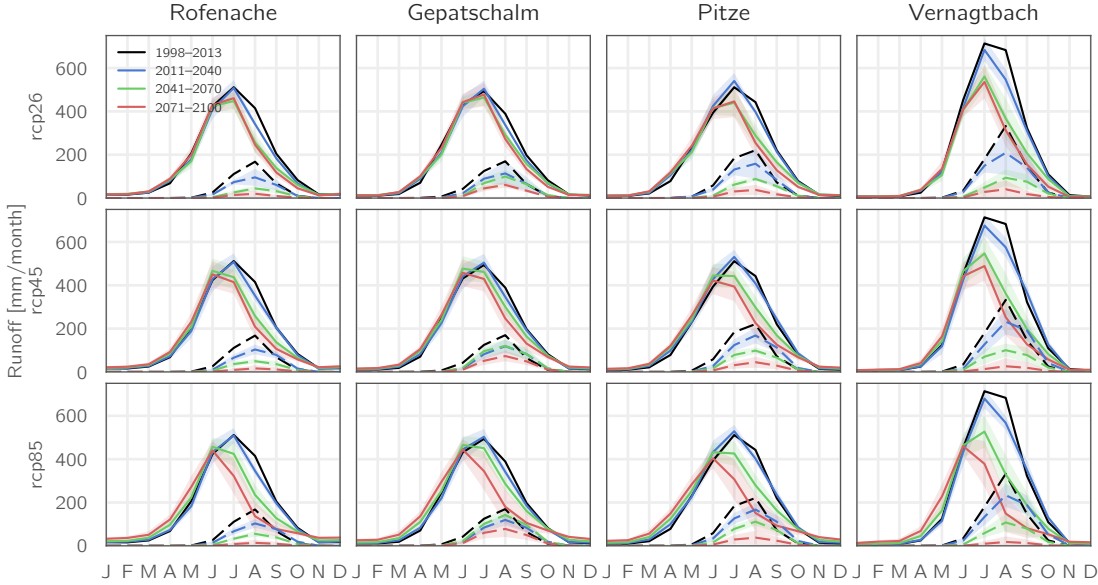

**Figure 15.** Average monthly runoff (multi-model mean $\pm$ one standard deviation indicated as shaded bands) as simulated for the early, middle, and late 21st century for four catchments and the three emission scenarios. Dashed lines indicate bare ice melt runoff.

hand only resulted in minor impacts on model performance. Figure 19 shows the average observed monthly runoff for the seven study catchments and the period 1998–2006, as well as the corresponding simulation results for the 50 m/1 h run using original hourly meteorological recordings and the 100 m/3 h run using disaggregated daily data, highlighting the tendency to overestimate runoff particularly during spring for the latter run. Table 3 shows the corresponding skill scores derived from

5 these two simulation runs: NSE, the benchmark efficiency (BE) (Schaefli and Gupta, 2007), and the percent bias (PBIAS). However, as for the scenario simulations mainly changes rather than absolute values are analyzed, these partial model biases likely do not affect the main conclusions of our study.

Overestimations of winter snow accumulation and, subsequently, glacier mass balances might partly also have consequences for (ii) the approach to account for glacier geometry changes. The $\Delta h$ parameterization as implemented is able to account only

10 for conditions of glacier retreat, while in case of positive mass balances no update of the glacier extents is performed. As Fig. 20 shows, 14–24 % of all simulated specific mass balances are positive in our simulations. However, as in previous studies the $\Delta h$ parameterization performed significantly better than alternative approaches such as accumulation area ratio-based methods and similarly well as complex ice flow models (Huss et al., 2010), we believe that this approach is an adequate tradeoff for the application on the regional scale where the application of of process-based ice flow models is not feasible.

15 We also assessed the influence of (iii) the initial glacier volume distribution by scaling the original initial ice thickness distribution by factors of 0.7 and 1.3, respectively (i. e., 30 % decrease/increase) and re-running the entire set of RCP4.5

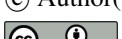



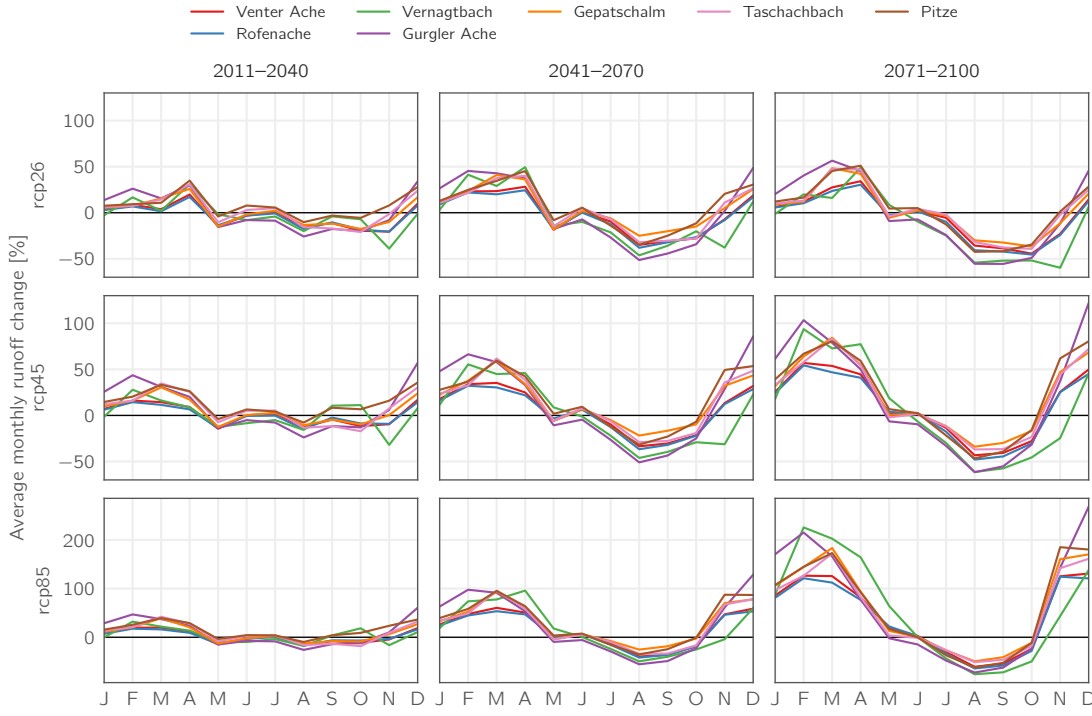

**Figure 16.** Average changes in monthly runoff relative to the historical period 1998–2013 over the course of the 21st century for the study catchments as simulated for the three emission scenarios (lines are multi-model means; note the differently scaled y-axes for RCP8.5).

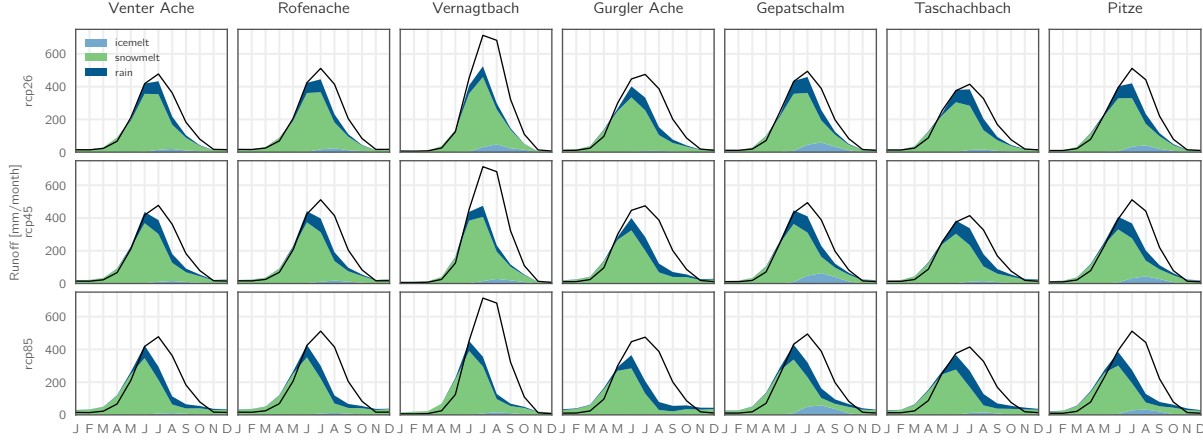

**Figure 17.** Average monthly runoff and composition (multi-model mean) for the period 2071–2100. The black lines show the average runoff in the historical period 1997–2013.





**Table 3.** NSE, BE, and PBIAS of observed vs. simulated runoff in the period 1998–2006 for the model runs using original hourly meteorological data and disaggregated daily data, respectively.

| ID | Catchment | Original | | | Disaggregated | | |
|----|-----------|-----|-----|-------------|-----|------|-------------|
|    |           | NSE | BE  | PBIAS [%]   | NSE | BE   | PBIAS [%]   |
| 1  | Venter Ache | 0.92 | 0.60 | 8.71 | 0.79 | -0.09 | 21.44 |
| 2  | Rofenache | 0.93 | 0.70 | 8.27 | 0.78 | 0.12 | 23.54 |
| 3  | Vernagtbach | 0.92 | 0.70 | -1.12 | 0.88 | 0.57 | 13.19 |
| 4  | Gurgler Ache | 0.91 | 0.59 | 1.96 | 0.83 | 0.22 | 18.87 |
| 5  | Gepatschalm | 0.94 | 0.68 | 3.09 | 0.88 | 0.41 | 11.95 |
| 6  | Taschachbach | 0.92 | 0.54 | 0.01 | 0.84 | 0.08 | 19.19 |
| 7  | Pitze | 0.94 | 0.72 | 2.30 | 0.88 | 0.45 | 16.46 |

simulations. However, as Fig. 21 (b–c) exemplarily shows for selected catchments, this does not significantly change the model results with regard to the evolution of glaciers and runoff volume. Similar results are obtained also for all other catchments.

With regard to (iv), many studies (e. g., Addor et al., 2014; Bosshard et al., 2013; Horton et al., 2006; Huss et al., 2014) have shown that the largest uncertainty in hydrological impact studies usually is due to the chosen climate models, and that

the spread between individual climate model simulations is often larger than the spread between different emission scenarios within a single climate model. In the case of GCM-RCM chains, GCMs tend to have a larger impact on the hydrological model results than the RCMs. We addressed these uncertainties by utilizing the entire range of available EURO-CORDEX simulations, resulting in a total of 14 GCM-RCM chains with five different GCMs (each driving 2–4 RCMs) and six different RCMs for the RCP4.5 and RCP8.5 scenarios. The results for the RCP2.6 scenario have to be interpreted with care, as for this

scenario only three GCM-RCM combinations (using two different GCMs) were available. Hence, especially when looking at multi-model averages, the RCP2.6 results are not directly comparable to the other two scenarios.

Besides the climate models themselves, also the spatial and temporal downscaling approaches from the coarse daily-resolution grids to the 3-hourly resolution point-scale time series add an uncertainty component to the results, as they are statistically derived from past conditions for which it is uncertain if they will persist in the future.

The influence of the climate models on the hydrological results is illustrated in Fig. 21 (a). Here, average monthly runoff for the period 2071–2100 as simulated by all 31 individual GCM-RCM realizations is shown exemplarily for the Pitze catchment. While the influence of the emission scenarios is visible to a part, there is significant overlap between the realizations of the different emission scenarios. For example, the realizations resulting in both the lowest and the highest August streamflow volumes are both driven by the RCP8.5 scenario. Similar results were already shown for the simulated glacier volume and area

in Fig. 11.




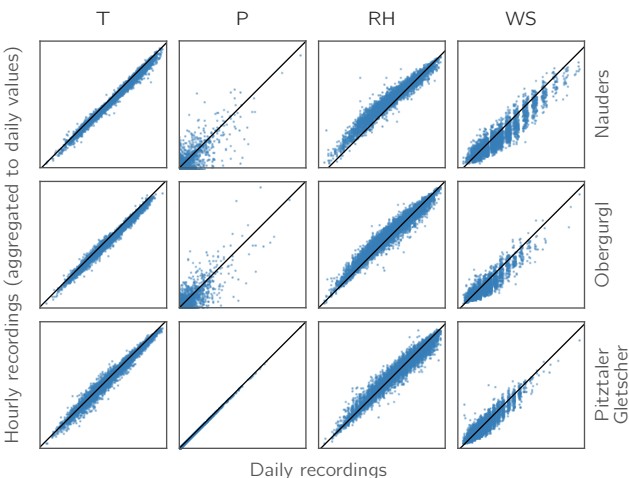

**Figure 18.** Scatter plots of daily vs. aggregated hourly recordings of air temperature (T), precipitation (P), relative humidity (RH), and wind speed (WS) for three stations in the study area.

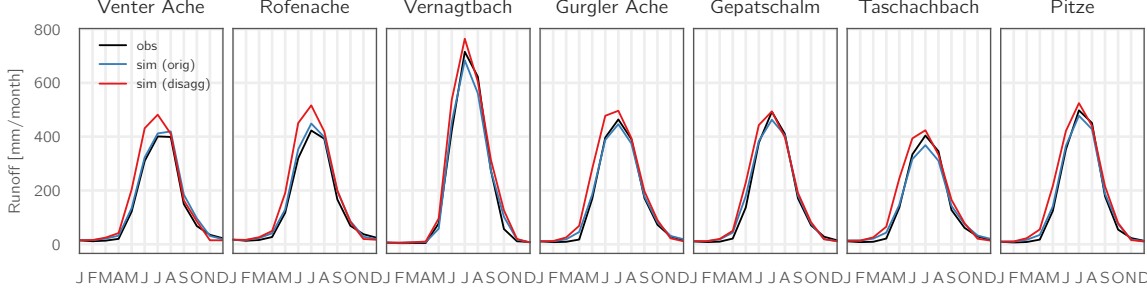

**Figure 19.** Seasonal cycle of observed (black lines) and simulated monthly runoff for the study catchments and the period 1998–2006. Blue lines correspond to the simulation results for the "reference run" (50 m spatial resolution, 1 h temporal resolution, original hourly meteorological data as forcing), while the red lines correspond to the simulation run using 100 m spatial and 3 h temporal resolution as well as disaggregated daily data as forcing.

## 5 Conclusions

In this study, we have forced a fully distributed physically based hydroclimatological model with the most state-of-the-art climate projections available. This is the most detailed study on cryospheric-hydrological climate change impacts in the Ötztal Alps to date, and to our knowledge also in high-elevation glacierized catchments in Austria in general.

5    While some uncertainty in the results is due to the model configuration, the largest uncertainty can be traced back to the climate projections. This leads to a considerable range in the projected snow coverage, glacier extents and hydrological regimes.





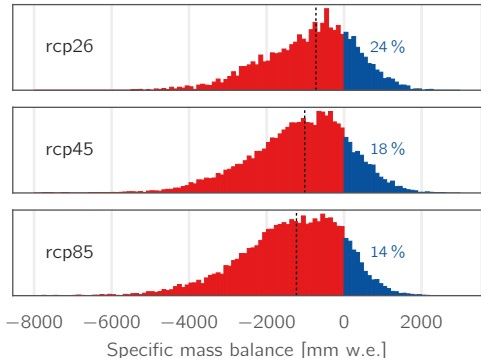

**Figure 20.** Histograms of the specific mass balance values as simulated for all (initially) 206 study area glaciers, all simulation years (2006–2100), and all GCM-RCM combinations. Dashed vertical lines refer to the median mass balance, and percentages indicate the total fractions of positive mass balances.

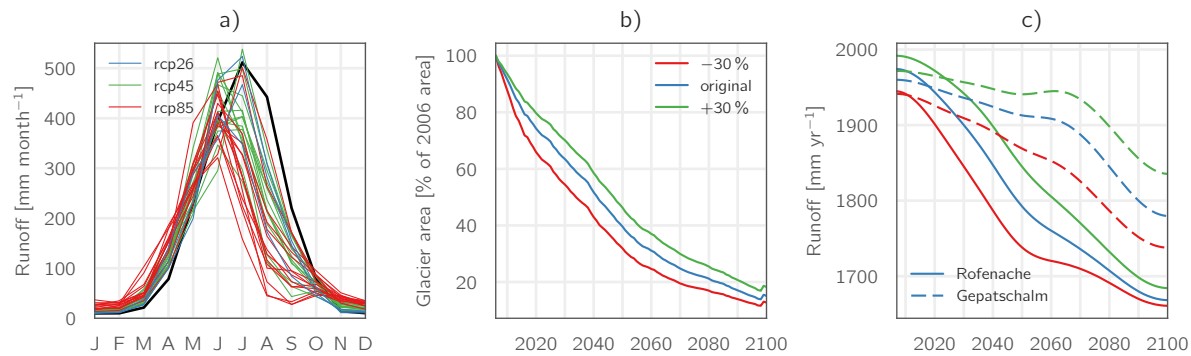

**Figure 21.** a) Average monthly runoff (period 2071–2100) for the Pitze catchment as simulated for all 31 individual GCM-RCM realizations, b) evolution of the total glacierized area for the entire Ötztal Alps study site as simulated by altering the initial ice thickness distribution by ±30 % (multi-model mean values for the RCP4.5 scenario), and c) corresponding runoff evolution (10-year Gaussian low-pass filtered) for the Rofenache and Gepatschalm catchments.

However, some common results can be found for all model runs. Snow cover is projected to decrease, however not as dramatic as presented in some other studies for Alpine regions due to the high elevation of the study site and strongly increasing winter precipitation, which partly compensates for the increased warming. Glaciers will continue to recede strongly throughout the century, and by 2100 most glaciers in the Ötztal Alps might have disappeared (depending on the considered emission scenario).

5 Resulting total glacierized area and glacier volume will likely amount to less than a quarter of today's state even for the RCP2.6 scenario. Consequently, glacier runoff will diminish proportionally and summer runoff will strongly decrease in all investigated catchments, resulting in a shift of the annual runoff peak from July towards June. Winter runoff volumes will




increase, however to still low absolute values. While the total annual runoff volumes stay approximately constant during the early 21st century compared to present-day levels, they gradually decrease throughout the rest of the century. Only for some catchments and scenarios runoff volumes slightly exceed present-day levels, indicating that the peak water period of maximum runoff is currently under way or has already passed.

5   *Competing interests.*   The authors declare that they have no conflict of interest.

*Acknowledgements.*   This work was carried out within the framework of the projects "W01 MUSICALS II – Multiscale Snow/Ice Melt Discharge Simulation into Alpine Reservoirs", carried out in the research programme of alpS – Centre for Climate Change Adaptation in Innsbruck, and "HydroGeM3", financed by the Austrian Academy of Sciences. The computational results presented have been achieved in part using the Vienna Scientific Cluster (VSC). The authors want to thank the COMET research programme of the Austrian Research Promotion Agency (FFG), the TIWAG – Tiroler Wasserkraft AG, and the Austrian Academy of Sciences. Meteorological and hydrological data were provided by the Zentralanstalt für Meteorologie und Geodynamik (ZAMG), the Hydrographic Service of Tyrol, the TIWAG – Tiroler Wasserkraft AG, the Commission for Glaciology of the Bavarian Academy of Sciences and Humanities, and the Autonomous Province of Bozen/Bolzano. We also would like to thank Matthias Huttenlau for managing the MUSICALS II project and many helpful discussions, and Johannes Schöber for valuable comments during the preparation of the manuscript.



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
