# Peer review of "Projected cryospheric and hydrological impacts of 21st century climate change in the Ötztal Alps (Austria) simulated using a physically based approach"

_Hydrology and Earth System Sciences, 2017_

## Referee Comment (RC1) · Anonymous Referee #1 · 19 Sep 2017

The authors present a very interesting and thorough study of the effects of a changing climate on cryospheric and hydrologic processes in the Austrian Alps over the next century. The authors use a well-established physically based model, which employs a full energy balance approach to simulate snow cover evolution, and a comprehensive future climate data set to look in detail at several processes in the Ötztal Alps. Additionally and commendably, the modelling method used in the study considers the change of glaciers in terms of ice thickness and areal extent throughout the modelling period. Results are presented for the development of the glaciers in the study area, the

changing depth and length of the snow covers, and for the amount and timing of the runoff in the area.

While comparable studies have been carried out in other parts of the European Alps (particularly Switzerland) this seems to be the first comprehensive study in the Eastern (Austrian Alps). This along with the already mentioned fact that the current study employs an energy balance model and includes glacier evolution in their analysis make this study very interesting for a larger readership. Overall, the paper is very well written. In total, it seems to be a bit on the long side. Maybe the authors could go over the paper again, trying to find some sections that could be shortened a bit. Especially the Method section seems to have some potential for shortening (in my opinion). Furthermore, some of the individual sentences seem to rather long which makes it a bit hard to read at times. It would probably be better to simply subdivide these sentences into two or three separate ones. Again I would encourage the authors to have one more look at the manuscript with this problem in mind.

The Abstract seems to me to be very concise while addressing all the major important points of the chosen method and the most significant results of the study. The introduction provides an adequate overview of previous studies in this field. The actual goals of this study are mentioned in one (rather short paragraph) at the end of the introduction. Here I would encourage the authors to maybe expand this part a little, explaining why this study area and this method was chosen and what sets this study apart from others before. The study site and data presentation are good and complete. The "Methods" section is very thorough, maybe even, as mentioned, a little on the long side with some potential for shortening. The "Results and Discussion" has a very logical setup and presents all relevant results clearly and concisely. Also included are appropriate comparisons of the results of the current studies to previous related studies. I especially applaud the authors for including an extensive section about the uncertainties in the modelling. Especially in modelling studies of the impact of future climate scenarios, such a section is highly valuable. Finally the "Conclusions" present the major results

of the study in a concise form. The conclusions are well based on the results and give the reader a good summary. The Tables and figures are adequate and present the information in an easily understandable form.

Overall, this is in my opinion a well thought out and well presented study. It covers a topic that is one of the most pressing questions of the coming decades not only in a scientific sense, but also for society as a whole (winter tourism, freshwater availability, etc.). The chosen geographic location, which to this date has not been studied in this context and the use of a completely physically based hydro-climatological model along with an algorithm tracking the evolution of the glaciers throughout the model period present, in my opinion, a significant new addition to the overall scientific knowledge. The paper's topic falls well within the scope of the journal and is of interest to other scientists, but also to the general public and political decision makers. I would therefore recommend publication after minor revisions. As noted above, these revisions could address the overall length of the paper and the sometimes excessively long sentences, as well as the following minor specific comments.

Specific Comments:

I'm guessing that the layout of the article is not the final version, but as it is now, some of the Figures are quite far away from where they are discussed in the text. This makes it somewhat hard to follow the discussion. I would make sure that the Figures are placed closer to the text discussion of them in the final version.

p.1 line 7-9: I would cut this sentence into two by putting a period after "century" and describing the situation below 1500m asl in a second sentence.

p.2 line 7-9: This sentence ("These general. . .") reads very awkward. Please try to rephrase.

p.4: The authors mention three climate scenarios. However, the scenarios are not explained any further until in the results section. Maybe you could add a small paragraph

explaining these secnarios i.e. what assumptions they make, how they are situated in the overall "scenario ensemble" (high, medium or low change assumptions)

p.6 line 15: Was terrain orientation and slope included in the calculation of the incoming solar radiation or were the model grid cells assumed to be "flat"?

p.6 p.32 Could you add one more sentence as to what modifications for climate conditions under forest canopies were applied?

p.7. Line 1: How was ground heat flux considered? Did you have ground temperatures (either measured or modelled) or was a constant value used?

p.8 line 20 and following: Is this paragraph really needed? It mainly discusses a method that is not needed for the current study. Maybe you could just explain what you did and omit the rest.

p. 13 line 19: "This is partly explainable from the fact" reads very awkward. Maybe replace with "This can be attributed to the fact"

Results Section: Generally, comparable studies start out by showing that the chosen model "system" start out by showing that the model system was able to simulate the current state (here 1997 to 2006) adequately. There is a Table (Table 3) much later, where some model efficiency values for the modelling of the current state are shown. However the focus of this Table is on showing original versus disaggregated values. I would welcome a short section showing how well the employed model system does for the historic phase at the start of the results section.

p.13 and 14: It is not entirely clear to me why observed data was used for the 1997 to 2006 period of the "regular" model runs, while RCM data was used for the 1971 – 2005 "snow cover model runs". Is this due to a lack of observed data from 1971 to 1997 or what is the reason for this procedure? Please explain.

p. 15 line 17: "Comparing" should be "Compared"

[Figure]

p. 15 line 30: End sentence after "century".

p.15 line 32: "However, also for the ...." is not a correct English sentence. Please rephrase.

p. 15 line 35: For "the" station Obergurgl....

p. 17 line29: "occur already" should be "already occur"

p.24 line 13: End sentence after (Huss et al 2010)

---

## Referee Comment (RC2) · Anonymous Referee #2 · 22 Oct 2017

PREMISE
The way I got to review this manuscript tells me one more time that the current peer-review system is close to collapse: I declined the invitation two times, and had no longer the courage of declining a third time, since I assumed that - at that stage - the desperation of the handling editor must have been larger than mine. Too many papers for too few reviewers seems to be at the heart of the problem. Here is a suggestion for a s simple fix: No decision shall be taken on any manuscript before the submitting group of authors has themselves provided three reviews (the typical number of reviews

for a manuscript) to some other articles. In the ideal world, these three reviews could be provided to any journal with decent quality, but since different publishers seem not to have any interest in collaborating between each other, the rule will probably be implementable only at the level of individual publishers. I may have overlooked some pitfalls that would go with such a suggestion, still, I'm convinced that it is worth a thought.

SUMMARY
Hanzer and colleagues present an analysis of future runoff evolution in a high-mountain catchment in the Ötztal Alps, Austria. Their analyses are based on the latest climate projections, i.e. in the results provided in the frame of the EURO-CORDEX initiative, and the physically-based model AMUNDSEN. The used methodology is sound, the paper is well written, the figure are illustrative, and the overall quality certainly adheres to HESS's standards. I have only series of minor remarks that the authors may or may not find useful to improve their work. Other than that, I look forward to see this contribution published soon.

SPECIFIC REMARKS
P1 L7ff: The abstract is well written. I was wondering, however, if the authors would be able to add half-a-sentence or so to better highlight the novelty in their contribution.
P3 L12ff: The wording seem to suggest that the methodology used by the authors is "better" that what has been used so far. Asked provocatively: How can the authors proof that? If such a proof is provided later in the manuscript, the authors may want to "announce" that here already.
P6 L14-15: Here, it was not clear to me how the authors will handle temperature lapse rate in simulations for the future (since it is said that the lapse rates are "calculated from point measurements"). One sentence of clarification could be helpful.
LP6 21ff: This is one of the very few points that I found conceptually problematic: The

authors use a "snow correction factor (SCF)" that, basically, increases the modelled amount of precipitation when precipitation is in its solid form. What does that mean for future simulations, i.e. for simulations in a warmer climate? I see the danger for the changes in precipitation to be overestimated: Since more precipitation will be liquid in future than it was in the past, less of the precipitation will be affected by the correction factor. With a correction factor chosen to be 1.15 (L198), my suspicion is that the effect could be as large as 15P7 L6ff: I think that the "SCF" (see above) should also be flagged as a parameter "requiring site-specific calibration".

P10 L3ff: I might have missed it later in the discussion, but here I thought that having at least one sentence addressing the limitation of quantile mapping methods would be appropriate. I refer to the limitations in handling extremes in particular.

P10 L9: Here I'm not sure to understand the wording "to retain intervariable relations". Maybe the authors can rephrase?

P10 L30: Several options for the number of considered grid-cells are named (1x1, 2x2,etc.). Which one was used at the end?

P12 L2-3: Here, the authors seems to additionally downscale the temporal resolution of the EURO-CORDEX results. Can a sentence be provided that explains why this is necessary?

P 12 L8-9: Still related to the above downscaling step: I found it rather problematic that the step apparently does not preserve the daily average temperature. Can the authors give a hint on how large the introduced deviations are, and whether these deviations are systematic? If so, an additional de-biasing step would seem appropriate.

P13 L6-7: I have difficulty in understanding the author's wording. Maybe they can rephrase?

P15 L17-21: The result that the Oetztaler Alps are projected to warm significantly less that the rest of the Alps seems an important one to me. Can the authors comment on whether this is likely to be a robust results, or whether it may just be caused by the comparison between studies using different methodologies? If the former (= robust result), a speculation on the causes could be very insightful.

P20 L11: The wording seems somewhat unfortunate to me: I wouldn't call a 54P20 L20ff: Different reactions in terms of runoff evolutions are noted for different sub-catchments. It would be useful if the authors could add some explanatory sentences for why this is the case.

P20 L34ff: The reported changes in winter runoff appear to be very large since they are expressed in P23 L21ff: I'm not entirely convinced about the "fairness" of the analysis investigating the effect of spatial and temporal resolution: Obviously, changing the resolution without re-calibrating the model will impact on model performance. The question for me would rather be about the changes in model performance once the model has been recalibrated. But maybe I simply misunderstood the authors' intentions.

P28 L5ff: Here (last part of the conclusions), I would have appreciated if some quantitative statements would have been included as well. Maybe, however, is just a matter of preferences. . .

Figure 4: I have difficulties in understanding why the median deviations for "G" and "WS" (including the full name of the variables in the figure caption would be very helpful!) are only positive. To me, this is an indication that the model debiasing is not working correctly (the mean deviation should be "zero" in that case).

---

## Author Comment (AC1) · 30 Nov 2017

*The authors present a very interesting and thorough study of the effects of a changing climate on cryospheric and hydrologic processes in the Austrian Alps over the next century. The authors use a well-established physically based model, which employs a full energy balance approach to simulate snow cover evolution, and a comprehensive future climate data set to look in detail at several processes in the Ötztal Alps. Additionally and commendably, the modelling method used in the study considers the change of glaciers in terms of ice thickness and areal extent throughout the modelling*

*period. Results are presented for the development of the glaciers in the study area, the changing depth and length of the snow covers, and for the amount and timing of the runoff in the area.*

*While comparable studies have been carried out in other parts of the European Alps (particularly Switzerland) this seems to be the first comprehensive study in the Eastern (Austrian Alps). This along with the already mentioned fact that the current study employs an energy balance model and includes glacier evolution in their analysis make this study very interesting for a larger readership. Overall, the paper is very well written. In total, it seems to be a bit on the long side. Maybe the authors could go over the paper again, trying to find some sections that could be shortened a bit. Especially the Method section seems to have some potential for shortening (in my opinion). Furthermore, some of the individual sentences seem to rather long which makes it a bit hard to read at times. It would probably be better to simply subdivide these sentences into two or three separate ones. Again I would encourage the authors to have one more look at the manuscript with this problem in mind.*

*The Abstract seems to me to be very concise while addressing all the major important points of the chosen method and the most significant results of the study. The introduction provides an adequate overview of previous studies in this field. The actual goals of this study are mentioned in one (rather short paragraph) at the end of the introduction. Here I would encourage the authors to maybe expand this part a little, explaining why this study area and this method was chosen and what sets this study apart from others before. The study site and data presentation are good and complete. The "Methods" section is very thorough, maybe even, as mentioned, a little on the long side with some potential for shortening. The "Results and Discussion" has a very logical setup and presents all relevant results clearly and concisely. Also included are appropriate comparisons of the results of the current studies to previous related studies. I especially applaud the authors for including an extensive section about the uncertainties in the modelling. Especially in modelling studies of the impact of future climate scenarios,*

*such a section is highly valuable. Finally the "Conclusions" present the major results of the study in a concise form. The conclusions are well based on the results and give the reader a good summary. The Tables and figures are adequate and present the information in an easily understandable form.*

*Overall, this is in my opinion a well thought out and well presented study. It covers a topic that is one of the most pressing questions of the coming decades not only in a scientific sense, but also for society as a whole (winter tourism, freshwater availability, etc.). The chosen geographic location, which to this date has not been studied in this context and the use of a completely physically based hydro-climatological model along with an algorithm tracking the evolution of the glaciers throughout the model period present, in my opinion, a significant new addition to the overall scientific knowledge. The paper's topic falls well within the scope of the journal and is of interest to other scientists, but also to the general public and political decision makers. I would therefore recommend publication after minor revisions. As noted above, these revisions could address the overall length of the paper and the sometimes excessively long sentences, as well as the following minor specific comments.*

We would like to thank the reviewer for their thorough evaluation of our manuscript and the very helpful and constructive comments, which are very much appreciated. With regard to your general comments, we agree that the paper is comparatively long and could benefit from some shortening. As a result of some of your comments and those of reviewer 2, we will remove some paragraphs in the revised version of the manuscript. In addition, we will expand on the goals and novelty of the study in the introduction. Please find below our replies to the individual comments.

*I'm guessing that the layout of the article is not the final version, but as it is now, some of the Figures are quite far away from where they are discussed in the text. This makes it somewhat hard to follow the discussion. I would make sure that the Figures are placed closer to the text discussion of them in the final version.*

We definitely agree that the placement of the figures in the discussion paper is unfortunate, and will make sure that this is improved for the final version.

*p.1 line 7-9: I would cut this sentence into two by putting a period after "century" and describing the situation below 1500m asl in a second sentence.*

We will change this sentence to: "Results show generally declining snow amounts with moderate decreases (0–20 % depending on the emission scenario) of mean annual snow water equivalent in high elevations (> 2500 m a.s.l.) until the end of the century. The largest decreases, amounting to up to 25–80 %, are projected to occur in elevations below 1500 m a.s.l."

*p.2 line 7-9: This sentence ("These general ... ") reads very awkward. Please try to rephrase.*

We will change this sentence to: "Since these projected impacts are mainly triggered by increasing temperatures (shift from snowfall to rainfall, earlier onset of snowmelt), the likelihood of occurrence is very high. In lower-elevated catchments on the other hand, projected changes in precipitation exhibit a larger impact on changes in runoff (Horton et al., 2006)."

*p.4: The authors mention three climate scenarios. However, the scenarios are not explained any further until in the results section. Maybe you could add a small paragraph explaining these secnarios i.e. what assumptions they make, how they are situated in the overall "scenario ensemble" (high, medium or low change assumptions)*

We will add a short paragraph explaining the main characteristics of the three RCPs to the respective section of the manuscript: "For the scenario simulations until 2100, we used the EURO-CORDEX climate change projections (Jacob et al., 2013) as climatic forcing, while considering the scenarios RCP2.6, RCP4.5, and RCP8.5. The latter is a scenario assuming no implementation of climate mitigation policies, resulting in considerably and steadily increasing emissions and concentrations of greenhouse gases over

time. RCP4.5 is an intermediate scenario consistent with peaking emissions around the mid-century and a strong decline afterwards, resulting in stabilizing concentrations by the end of the century. Finally, the intervention scenario RCP2.6 is at the very low end of the spectrum in terms of future emissions and radiative forcing, assuming a peak in $CO_2$ concentrations in the middle of the century and a slow decline afterwards along with negative emissions toward the end of the century."

*p.6 line 15: Was terrain orientation and slope included in the calculation of the incoming solar radiation or were the model grid cells assumed to be "flat"?*

Yes, in the radiation model for each grid cell a unit vector normal to the surface is calculated which is the basis for the subsequent calculations. We will add this information to the manuscript and change P6 L15f. to "(...) while taking into account terrain slope and orientation, hill shading, transmission losses and gains due to scattering, absorption, and reflections (...)".

*p.6 p.32 Could you add one more sentence as to what modifications for climate conditions under forest canopies were applied?*

We will change the sentence to: "In forested areas the interpolated meteorological fields are modified in order to capture sub-canopy conditions, resulting in reduced shortwave radiation, precipitation, and wind speed, increased longwave radiation and humidity, and an attenuation of the diurnal temperature cycle. Additionally the effects of the forest snow processes of interception, sublimation, and melt unload are accounted for (Strasser et al., 2011)."

*p.7. Line 1: How was ground heat flux considered? Did you have ground temperatures (either measured or modelled) or was a constant value used?*

Ground heat flux was assumed to be constant at 2 W/m$^2$.

*p.8 line 20 and following: Is this paragraph really needed? It mainly discusses a method that is not needed for the current study. Maybe you could just explain what you did and*

*omit the rest.*

We agree. We will remove this paragraph from the manuscript.

*p. 13 line 19: "This is partly explainable from the fact" reads very awkward. Maybe replace with "This can be attributed to the fact"*

Done.

*Results Section: Generally, comparable studies start out by showing that the chosen model "system" start out by showing that the model system was able to simulate the current state (here 1997 to 2006) adequately. There is a Table (Table 3) much later, where some model efficiency values for the modelling of the current state are shown. However the focus of this Table is on showing original versus disaggregated values. I would welcome a short section showing how well the employed model system does for the historic phase at the start of the results section.*

The validation of the model setup for historical conditions has been described in detail in a previous paper (Hanzer et al., 2016), which is why we generally refer to this reference and only concisely discuss the changes in model performance regarding the few changes in model setup (reduced spatial and temporal resolution, temporal disaggregation of forcing data) in section 4.5. We agree that this might not be immediately clear from the article and will add a short paragraph for clarification to the start of the results section.

*p.13 and 14: It is not entirely clear to me why observed data was used for the 1997 to 2006 period of the "regular" model runs, while RCM data was used for the 1971 – 2005 "snow cover model runs". Is this due to a lack of observed data from 1971 to 1997 or what is the reason for this procedure? Please explain.*

Yes, the reason for using the historical RCM data instead of observed station data was to avoid introducing inhomogeneities in the simulation results due to the highly varying temporal coverage of station observations in the past. By using the historical RCM

simulations and applying the same downscaling and bias correction methodology as for the future period, homogeneous and gap-free time series for the period 1971–2100 are obtained for all stations and variables. The model runs using observed data on the other hand were used (i) for validation, (ii) for deriving the ice thickness distribution for 2006 used for initializing the scenario runs, and (iii) as reference for calculating the future changes in runoff (figs. 15–17). We will clarify this also in the manuscript.

*p. 15 line 17: "Comparing" should be "Compared"*

Done.

*p. 15 line 30: End sentence after "century".*

Done.

*p.15 line 32: "However, also for the ... ." is not a correct English sentence. Please rephrase.*

We will change this sentence to: "For the other two scenarios comparatively small decreases in average SWE are simulated in this period, amounting to 427 mm ($-18$ %) for RCP4.5 and 357 mm ($-31$ %) for RCP8.5."

*p. 15 line 35: For "the" station Obergurgl ... .*

Done.

*p. 17 line29: "occur already" should be "already occur"*

Done.

*p.24 line 13: End sentence after (Huss et al 2010)*

Done.

**References**

Hanzer, F., Helfricht, K., Marke, T., Strasser, U. (2016). Multilevel spatiotemporal

validation of snow/ice mass balance and runoff modeling in glacierized catchments. The Cryosphere, 10(4), 1859–1881. http://doi.org/10.5194/tc-10-1859-2016

Jacob, D., Petersen, J., Eggert, B., Alias, A., Christensen, O. B., Bouwer, L. M., et al. (2013). EURO-CORDEX: new high-resolution climate change projections for European impact research. Regional Environmental Change, 14(2), 563–578. http://doi.org/10.1007/s10113-013-0499-2
* * *

---

## Author Comment (AC2) · 30 Nov 2017

*PREMISE*

*The way I got to review this manuscript tells me one more time that the current peer-review system is close to collapse: I declined the invitation two times, and had no longer the courage of declining a third time, since I assumed that - at that stage - the desperation of the handling editor must have been larger than mine. Too many papers for too few reviewers seems to be at the heart of the problem. Here is a suggestion*
*for a s simple fix: No decision shall be taken on any manuscript before the submitting group of authors has themselves provided three reviews (the typical number of reviews for a manuscript) to some other articles. In the ideal world, these three reviews could be provided to any journal with decent quality, but since different publishers seem not to have any interest in collaborating between each other, the rule will probably be implementable only at the level of individual publishers. I may have overlooked some pitfalls that would go with such a suggestion, still, I'm convinced that it is worth a thought.*

*SUMMARY*

*Hanzer and colleagues present an analysis of future runoff evolution in a high- mountain catchment in the Ötztal Alps, Austria. Their analyses are based on the latest climate projections, i.e. in the results provided in the frame of the EURO-CORDEX initiative, and the physically-based model AMUNDSEN. The used methodology is sound, the paper is well written, the figure are illustrative, and the overall quality certainly adheres to HESS's standards. I have only series of minor remarks that the authors may or may not find useful to improve their work. Other than that, I look forward to see this contribution published soon.*

First of all, we wish to express that we share the reviewer's concerns regarding the current state of the peer review system. While it will likely not be an easy task to fix this problem, we wish to thank you for your valuable thoughts that we will take over as good ideas to be thoroughly considered in the community. Most importantly, we are thankful that you nevertheless took the time to review our manuscript and for the many helpful and constructive comments. Please find below our replies to the individual comments.

*P1 L7ff: The abstract is well written. I was wondering, however, if the authors would be able to add half-a-sentence or so to better highlight the novelty in their contribution.*

We will add the following sentence to the abstract to highlight the novelties in our work: "The high level of process representation within the model, the high spatial and temporal model resolution, and the large number and range of considered climate model

runs make these findings a novel contribution to the possible impacts of future climate change in the Ötztal Alps in particular and in high-elevation Alpine catchments in general."

*P3 L12ff: The wording seem to suggest that the methodology used by the authors is "better" that what has been used so far. Asked provocatively: How can the authors proof that? If such a proof is provided later in the manuscript, the authors may want to "announce" that here already.*

We certainly did not intend to imply that our methodology is "better" than other approaches. While we state in the introduction that more physically based models are potentially better suited for the application under changing (climatic) conditions than simple conceptual models, in the study we emphasize also the limitations of our methodology and attempt to quantify the uncertainties in the results. To follow up on this, in a future study we will compare our results with those obtained from applying a semi-distributed conceptual model in the same region and using the same forcing data, which will allow to investigate the uncertainties induced by different modeling approaches in more detail.

*P6 L14-15: Here, it was not clear to me how the authors will handle temperature lapse rate in simulations for the future (since it is said that the lapse rates are "calculated from point measurements"). One sentence of clarification could be helpful.*

For the application in this study, temperature lapse rates were not calculated dynamically but rather prescribed in the form of static monthly values which do not change in the future. In the case of dynamic calculation, the lapse rates are calculated separately for each time step by regressing the point data (which can be either actual measurements or e.g. downscaled climate model data) against elevation. We agree that the term "point measurements" in P6 L15 is too specific and will replace it by "point data".

*P6 21ff: This is one of the very few points that I found conceptually problematic: The authors use a "snow correction factor (SCF)" that, basically, increases the modelled*

*amount of precipitation when precipitation is in its solid form. What does that mean for future simulations, i.e. for simulations in a warmer climate? I see the danger for the changes in precipitation to be overestimated: Since more precipitation will be liquid in future than it was in the past, less of the precipitation will be affected by the correction factor. With a correction factor chosen to be 1.15 (L198), my suspicion is that the effect could be as large as 15*

We agree that this is a valid concern. The increase in snowfall amounts by 15 % (which is applied additionally to the wind speed and temperature-dependent precipitation adjustment) is an empirical correction, however this value has been derived using a very thorough validation procedure taking into account various validation data sets such as areal precipitation, point-based snow depths, lidar-derived snow depth maps, and multi-year glacier mass balances (Hanzer et al., 2016). Although it has been shown that it is absolutely necessary to correct the observed precipitation amounts for undercatch in the study region, we agree that this fixed value might change given that climate conditions change as well. Utilizing the RCM-simulated snowfall fractions and precipitation fields instead of downscaling to point locations could reduce this uncertainty. Promising advances in downscaling methods such as quasi-dynamical approaches as e.g. in the ICAR model (Gutmann et al., 2016) could bridge this gap between statistical and dynamical methods and allow deriving more realistic small-scale precipitation fields. While this was not feasible for the present study, the revised version of the manuscript will include a short discussion on the possible uncertainties in model results induced by the precipitation downscaling and snow correction approach.

*P7 L6ff: I think that the "SCF" (see above) should also be flagged as a parameter "requiring site-specific calibration".*

You are right. We will change this sentence to: "Apart from the parameters of this linear reservoir model which have to be calibrated individually for each catchment, most parameters in the model have a physical meaning, and in general no site-specific calibration beyond the parameters for the runoff module and the precipitation correction
method is performed."

*P10 L3ff: I might have missed it later in the discussion, but here I thought that having at least one sentence addressing the limitation of quantile mapping methods would be appropriate. I refer to the limitations in handling extremes in particular.*

We agree. We will briefly expand on the limitations of QM in the respective section of the article.

*P10 L9: Here I'm not sure to understand the wording "to retain intervariable relations". Maybe the authors can rephrase?*

We will change the wording to: "(. . .) and can preserve intervariable dependency structures".

*P10 L30: Several options for the number of considered grid-cells are named (1x1, 2x2,etc.). Which one was used at the end?*

The number of grid cells used for the averaging is not statically defined, but rather determined dynamically for each model, variable, and station using the methodology described in section 3.3 (P11 L1–6). As seen in fig. 4 of the manuscript, a value of 1x1 is the most common one, however all values of up to 10x10 are occurring.

*P12 L2-3: Here, the authors seems to additionally downscale the temporal resolution of the EURO-CORDEX results. Can a sentence be provided that explains why this is necessary?*

We will change the first sentence of section 3.4 to: "For the calculation of the snow and ice surface energy balance in AMUNDSEN, 1–3-hourly meteorological input time series are required in order to capture the diurnal variability of the contributing energy fluxes. As the EURO-CORDEX simulations were however only available in daily temporal resolution, an additional processing step was necessary."

*P 12 L8-9: Still related to the above downscaling step: I found it rather problematic that*

*the step apparently does not preserve the daily average temperature. Can the authors give a hint on how large the introduced deviations are, and whether these deviations are systematic? If so, an additional de-biasing step would seem appropriate.*

We agree that ideally the daily mean temperature should be preserved, however given daily values of Tmin, Tmax, and Tmean, with the implemented disaggregation schemes (either assuming a sinusoidal temperature course or an "average" temperature course derived from hourly observations) it is possible to preserve only two of them (either Tmin and Tmax, or Tmean and (Tmax – Tmin)). Preserving Tmin and Tmax in combination with the other chosen disaggregation methods yielded better results with respect to the multilevel validation of the AMUNDSEN model results for the past, hence this method was used. The mean absolute errors of hourly observed vs. disaggregated temperature values are similar for both methods (1.12 °C for preserving Tmean and 1.19 °C for preserving Tmin and Tmax (values are averages over all stations)), however preserving Tmin and Tmax leads to a general tendency towards underestimated temperatures (0.38 °C on average).

*P13 L6-7: I have difficulty in understanding the author's wording. Maybe they can rephrase?*

We will rephrase P13 L5-7 to: "For relative humidity, an additional disaggregation method based on Waichler and Wigmosta (2003) was implemented. Hourly humidity values are generated using [month, hour, dry/wet day] categorical mean values with the additional option to preserve the daily mean humidity."

*P15 L17-21: The result that the Oetztaler Alps are projected to warm significantly less that the rest of the Alps seems an important one to me. Can the authors comment on whether this is likely to be a robust results, or whether it may just be caused by the comparison between studies using different methodologies? If the former (= robust result), a speculation on the causes could be very insightful.*

Thank you for pointing this out. In fact, further analyses show that this result can likely

be attributed to the different methodologies used in the individual studies (comparison of downscaled and bias-corrected RCM data interpolated to a 100 m grid (our study) vs. raw RCM results (the other studies)). A robust comparison of future climate change in our study region vs. the Alps would need to be conducted using data sets generated with the same or comparable methodology. As this paragraph is not essential for our study, we will remove it in the revised version of the manuscript.

*P20 L11: The wording seems somewhat unfortunate to me: I wouldn't call a 54*

Unfortunately, parts of your comment seem to be missing, however likely you argue that a 54 % reduction in glacier area cannot be called "moderate", to which we agree. We will remove "comparatively moderate".

*P20L20ff: Different reactions in terms of runoff evolutions are noted for different sub-catchments. It would be useful if the authors could add some explanatory sentences for why this is the case.*

The differences are mostly due to differences in glacierization and total ice volume/average ice thickness in the respective catchments. We will add some explanatory remarks to the respective section of the manuscript.

*P20 L34ff: The reported changes in winter runoff appear to be very large since they are expressed in*

Indeed, the relative changes in winter runoff are very large however still corresponding to very low absolute values, which we also emphasize in the manuscript. Unfortunately, here again parts of your comment seem to have been cut off, hence we would politely ask to resend it if you suggest making changes to this part of the manuscript.

*P23 L21ff: I'm not entirely convinced about the "fairness" of the analysis investigating the effect of spatial and temporal resolution: Obviously, changing the resolution without re-calibrating the model will impact on model performance. The question for me would rather be about the changes in model performance once the model has been*

*recalibrated. But maybe I simply misunderstood the authors' intentions.*

In the model, it is generally aimed to reduce the need for calibration. As stated above, the only parameters that have to be calibrated are the linear reservoir coefficients and possibly the snow correction factors. The former were in fact recalibrated for the assessment of the changes in model performance due to the changes in resolution and forcing data, however in the case of the latter the SCF of 15 % used in the original model setup (in combination with the temperature and wind speed-based correction and the openness-based snow redistribution) still yielded the best overall results with respect to the multilevel validation procedure. Hence, we believe our conclusion that model performance is not majorly affected by the changes in temporal and spatial resolution is appropriate.

*P28 L5ff: Here (last part of the conclusions), I would have appreciated if some quantitative statements would have been included as well. Maybe, however, is just a matter of preferences ...*

We agree that some quantitative statements would be appropriate here. We will replace the respective section of the conclusions by: "While some uncertainty in the results is due to the model configuration, the largest uncertainty can be traced back to the climate projections. This leads to a considerable range in the projected snow coverage, glacier extents and hydrological regimes. Snow cover is projected to decrease by up to 80 % in elevations below 1500 m a.s.l., while only comparatively moderate decreases (up to 25 %) are found for high-elevated areas (> 2500 m a.s.l.) due to strongly increasing winter precipitation which partly compensates for the increased warming. Glaciers will continue to recede strongly throughout the century. Until 2050, glacier volume will decline by approx. 60–65 % largely independent of the emission scenario, whereas by the end of the century 80–96 % of the original ice volume will be lost. Consequently, glacier runoff will diminish proportionally and summer runoff will strongly decrease in all investigated catchments by up to 55 %, resulting in a shift of the annual runoff peak from July towards June. Winter runoff volumes on the other hand will increase, however

to still low absolute values. While the total annual runoff volumes stay approximately constant during the early 21st century compared to present-day levels, they gradually decrease throughout the rest of the century. Only for some catchments and scenarios runoff volumes slightly exceed present-day levels, indicating that the peak water period of maximum runoff is currently under way or has already passed in this region."

*Figure 4: I have difficulties in understanding why the median deviations for "G" and "WS" (including the full name of the variables in the figure caption would be very helpful!) are only positive. To me, this is an indication that the model debiasing is not working correctly (the mean deviation should be "zero" in that case).*

Global radiation is the variable with the least amount of available stations (3). For two of these stations, the mean deviation (MD) of bias-corrected vs. observed values is (absolutely) less than 0.025 $W/m^2$ for all models, whereas for one high-altitude station (Vernagtbach, 2640 m a.s.l.) it amounts to up to 1.1 $W/m^2$ (mean: 0.80 $W/m^2$), which however corresponds to a deviation of only approx. 0.5 percent in relative terms. Similarly, bias-corrected wind speed values partly are very slightly positively biased, however amounting to a maximum MD of 0.08 m/s over all models and stations. Hence, we believe these very small remaining biases are negligible for our subsequent analyses.

**References**

Gutmann, E., Barstad, I., Clark, M., Arnold, J., Rasmussen, R. (2016). The Intermediate Complexity Atmospheric Research Model (ICAR). Journal of Hydrometeorology, 17(3), 957–973. http://doi.org/10.1175/JHM-D-15-0155.1

Hanzer, F., Helfricht, K., Marke, T., Strasser, U. (2016). Multilevel spatiotemporal validation of snow/ice mass balance and runoff modeling in glacierized catchments. The Cryosphere, 10(4), 1859–1881. http://doi.org/10.5194/tc-10-1859-2016

Waichler, S. R., Wigmosta, M. S. (2003). Development of Hourly Meteorological Values From Daily Data and Significance to Hydrological Modeling at H.

J. Andrews Experimental Forest. Journal of Hydrometeorology, 4(2), 251–263. http://doi.org/10.1175/1525-7541(2003)4<251:DOHMVF>2.0.CO;2